# The predictability of reported drought events and impacts in the Ebro Basin using six different remote sensing data sets

Clara Linés[1], Micha Werner[1,2], and Wim Bastiaanssen[1,3]

[1]IHE Delft, Department of Water Science and Engineering, 2601 DA, Delft, the Netherlands
[2]Deltares, 2600 MH, Delft, the Netherlands
[3]Delft University of Technology, Faculty of Civil Engineering and Geosciences, 2628 CD, Delft, the Netherlands

*Correspondence to:* C. Linés (c.lines@un-ihe.org)

**Abstract.** The implementation of drought management plans contributes to reduce the wide range of adverse impacts caused by water shortage. A crucial element of the development of drought management plans is the selection of appropriate indicators and their associated thresholds to detect drought events and monitor the evolution. Drought indicators should be able to detect emerging drought processes that will lead to impacts with sufficient anticipation to allow measures to be undertaken effectively.
However, in the selection of appropriate drought indicators the connection to the final impacts is often disregarded. This paper explores the utility of remotely sensed data sets to detect early stages of drought at the river basin scale, and how much time can be gained to inform operational land and water management practices. Six different remote sensing data sets with different spectral origin and measurement frequency are considered, complemented by a group of classical in situ hydrologic indicators. Their predictive power to detect past drought events is tested in the Ebro basin. Qualitative (binary information based on media records) and quantitative (crop yields) data of drought events and impacts spanning a period of 12 years are
used as a benchmark in the analysis. Results show that early signs of drought impacts can be detected up to some 6 months before impacts are reported in newspapers, with the best correlation-anticipation relationships for the Standard Precipitation Index (SPI), the Normalized Difference Vegetation Index (NDVI) and Evapotranspiration (ET). Soil Moisture (SM) and Land Surface Temperature (LST) offer also good anticipation, but with weaker correlations, while Gross Primary Production (GPP)
presents moderate positive correlations only for some of the rainfed areas. Although classical hydrological information from water levels and water flows provided better anticipation than remote sensing indicators in most of the areas, correlations were found to be weaker. The indicators show a consistent behaviour with respect to the different levels of crop yield in rainfed areas among the analysed years, with SPI, NDVI and ET providing again the stronger correlations. Overall, the results confirm remote sensing products' ability to anticipate reported drought impacts and therefore appear as a useful source of information
to support drought management decisions.

## 1  Introduction

Drought is defined as a temporary water shortage in part caused by anomalous climatic conditions but strongly influenced by socio-economic factors (Kallis, 2008). The effects of drought propagate through all human and natural systems that depend on water directly or indirectly producing substantial losses (Wilhite et al., 2007). Various economic sectors are adversely

affected, in particular agricultural production, energy generation, and water supply for domestic and industrial use. Habitat degradation, increased mortality of flora and fauna, and increased occurrence of wildfires are examples of the effects on the natural environment. Indirect impacts, such as increase on prices, unemployment or migration, arise as a consequence of the direct impacts, and may be felt in a much wider area, even reaching the global scale (Wilhite and Vanyarkho, 2000).

The occurrence and severity of drought impacts depends on the intensity and duration of the event, but also on the vulnerability of the society and the environment (Wilhite, 2000). As a consequence the conditions that produce negative socio-economic impacts are not necessarily the same for the different sectors that may be affected (Redmond, 2002). The timing of the event also influences the severity of impacts. Soil moisture deficit during the flowering stage of a crop or reduced domestic water supplies during the tourist season are examples of situations in which the socio-economic impact is aggravated due to the
timing of the drought. The aim of the current paper is to identify earth observation data sets that can be used to detect early stages of drought at the basin scale, as well as to what extent these data sets can anticipate drought impacts, and be used to inform operational land and water management.

The implementation of drought management plans by governing agencies can contribute to reducing the negative effects of drought by guiding decision makers in taking appropriate mitigation actions. However, the effectiveness and cost-efficiency
of these actions rely on the selection of suitable indicators to monitor drought conditions, and to detect events at an early stage, gaining valuable time for mitigation measures to be implemented effectively and impacts to be mitigated. Examples of actions that can be taken include retention of water; reallocation of available water resources; curtailment of current allocations; recommendations to plant less water-demanding or drought resistant crops; or prohibition of certain water uses (e.g. watering gardens or washing cars).

Indicator systems consist of drought indices with associated thresholds that allow classifying the event in categories of drought severity. A classical examples is the division of river flow into several categories. When the value of the indicator crosses one of the thresholds, managers should decide whether to activate the corresponding responses defined in the drought management plan for that situation. Indicators and associated thresholds should be problem, context and user-specific (Kallis, 2008), and therefore an integrated management of droughts in basins where there are multiple users requires advanced drought
detection systems based on multiple indicators.

Measurements from in situ networks and from remote sensing are complementary sources that can be used to build the system of indicators for early detection and monitoring of drought conditions. In situ data is generally collected at specific points only. The advantage is the high temporal frequency of observations, and the availability of longer term records. Remote sensing techniques, on the other hand, offer cost effective and spatially continuous information over extended regions. Satellites
allow drought events to be categorised over a certain area, rather than at point locations (Kogan, 2001; Famiglietti et al., 2015). Several satellite data sets are now available at a daily or at even shorter time scales, offering excellent potential to develop sound drought monitoring systems in real time, and allowing to overcome the shortcomings of classical indicators based on in situ data sets that lack the spatial scale (e.g. Sheffield et al., 2014; van Dijk and Renzullo, 2011).

Keyantash and Dracup (2002) analyse a set of criteria to assess the usefulness of drought indicators for the assessment of
drought severity and point out that while the robustness of an indicator provides insight into its consistent behaviour across

differing conditions, assessing the accuracy of the information provided by the indicator requires a standard or benchmark for comparison. This holds true for both remote sensing-based and ground-based indicators. A standard that offers an absolute metric of drought is not easily available and likely does not exist, and as a result a common approach to evaluating the performance of remote sensing-based drought indicators is to assess their robustness by comparing them with other indicators such as flow, reservoir levels or widely used drought indices (e.g. Morid et al., 2006; Tsakiris et al., 2006; Vasiliades et al., 2011). Expert knowledge may also be used in practical applications as a benchmark to assess drought indicators such as in Steinemann et al. (2015), who rely on regional water managers, drought decision-makers and other stakeholders' knowledge as a reference to develop, select and evaluate drought indicators. Expert judgement is also included, in combination with several indicators and model outputs, in the U.S. drought monitor (Svoboda et al., 2002) to develop a weekly map of drought conditions in the U.S. which itself is also frequently selected as a reference data set in the evaluation of the performance of drought indicators in the country (e.g. Anderson et al., 2011, 2013; Brown et al., 2008).

Since mitigating impacts is the purpose of drought indicators included in drought management strategies, impact data is especially suitable as a benchmark in this case. Drought impacts, however, are difficult to evaluate and are rarely monitored (Wilhite, 2011; Lackstrom et al., 2013). Several studies have analysed the connections of drought indices to quantifiable effects on agriculture, hydrology or forests (see Vicente-Serrano et al., 2012, for a review), but very few have applied the impact data (mostly crop yields) as a benchmark to assess indicators for drought detection (e.g. Potop, 2011; Sepulcre et al., 2012; Stagge et al., 2015). Recognising the potential of this kind of data for drought management, two large scale initiatives have recently been launched: the US Drought Impact Reporter (DIR) (Wilhite et al., 2007) and the European Drought Impact report Inventory (EDII) (Stahl et al., 2016). These have the objective to collect text-based impact records systematically with the aim to increase their availability and accessibility. Recent studies have explored the links of the EDII records to drought indicators (Bachmair et al., 2015, 2016; Blauhut et al., 2015), though these have focused on the national scale, and it is recognised that further development is required to allow analysis at the subnational scale (Stahl et al., 2016).

Despite their important role in mitigation of drought impacts, the selection and use of indicators and thresholds for decision making often suffers from a lack of scientific justification: only a few studies have analysed the choice of drought indicators in relation to drought management in practice (Steinemann and Cavalcanti, 2006; Steinemann et al., 2015). Moreover, the thresholds that have been selected to declare droughts are only rarely connected to the specific impacts that need to be avoided (Wilhite, 2000). In this paper quantitative and qualitative drought impact information are applied as a benchmark in evaluating the utility of indicators derived from six different remote sensing data sets at river basin scale. This implies that the analysis is not based on a definition of drought as a statistical extreme, but as the occurrence of certain conditions of meteorological origin that lead to impacts in sectors depending on water. Two aspects are considered in the assessment of the indicators against the benchmark data: how well do these indicators reflect reported drought impacts and to what degree can these indicators be used to anticipate drought conditions and consequent impacts.

## 2 Material and methods

### 2.1 The Ebro basin

The Ebro basin, with an extent of 85,600 km², is the largest catchment in Spain. It is located in the north-east, bounded by the Pyrenees and Cantabrian mountain ranges to the north, and the Iberian System to the south. It is a highly regulated basin with 51 reservoirs ($> 1$ Mm$^3$) and a total storage capacity of more than 7,500 Mm$^3$, which supply water to more than 900,000 has of irrigated agriculture and more than 450 hydroelectrical plants (CHE, n.d.). Analysis of the impacts of a recent drought event (2005–2008) revealed that agriculture and food production are the main sectors affected by drought in the area, but impacts to hydropower production, water supply to villages, food industry, recreational activities and ecosystem functions were also identified (Perez y Perez and Barreiro-Hurlé, 2009; Hernández-Mora et al., 2013).

The period 2000-2012, selected for the analysis, encompasses a wide range of different conditions: the hydrological year 2004–2005 was characterised as one of the most intense droughts of the record in the Iberian Peninsula (García-Herrera et al., 2007), while 2003–2004 is considered one of the wettest hydrological years of the country's record (MMA, 2005).

The Confederación Hidrográfica del Ebro (CHE) is the organization responsible for the management, regulation and conservation of water in the Ebro basin. The basin is divided into 18 management units, each of which has a board constituted of representatives of the different water users as well as of the basin authority to coordinate the use of the hydraulic infrastructures and water resources in their area.

A drought management plan for the basin was developed in 2007 to guide drought management actions (CHE, 2007). The plan defines a set of indicators to detect situations of hydrological drought in the Ebro basin and evaluate their severity. The indicators are built using observations from a rich network of in situ automatic stations. In the areas in which the flow is regulated by dams, water stored in reservoirs is considered the most robust indicator, but other variables such as water flow, snow depths or head levels in aquifers may also be taken into account. For areas with a natural or an almost natural flow regime without reservoirs, the three-month water flow measured at representative stations is selected as the main indicator. In one of the management units, where there is no regulation and no representative rivers, groundwater levels are used as indicator.

The north-east of the basin, where the larger irrigation districts of the Ebro basin are located, was selected to evaluate the set of drought indicators against the qualitative text reports (Figure 1). This area was also the most affected by the drought period 2005–2006 (Hernández-Mora et al., 2013). It is composed of four management units (management units 12 to 15). Figure 1 shows the management units further subdivided according to the main drought indicators currently selected in each: three-month water flow in the northern sectors (zones 120, 130, 140 and 150) and reservoir levels in the southern sectors. To differentiate non irrigated agricultural areas, Corine Land Cover map 2006 (CLC06) classes 211 (non-irrigated arable land), 242 (complex cultivation patterns) and 231 (pastures) have been used. The land cover class *irrigated agriculture* corresponds to the irrigation polygons provided by MAGRAMA (1997 version). The main irrigation districts are marked with dotted patterns and are identified by the name of the main canal that serves them. For the evaluation of quantitative crop yield data as a benchmark, only five of the districts within this area were selected: Hoya Huesca (H), Somontano (S), La Litera (L), Monegros (M), and Bajo Cinca (B), including irrigated and rainfed cropland.

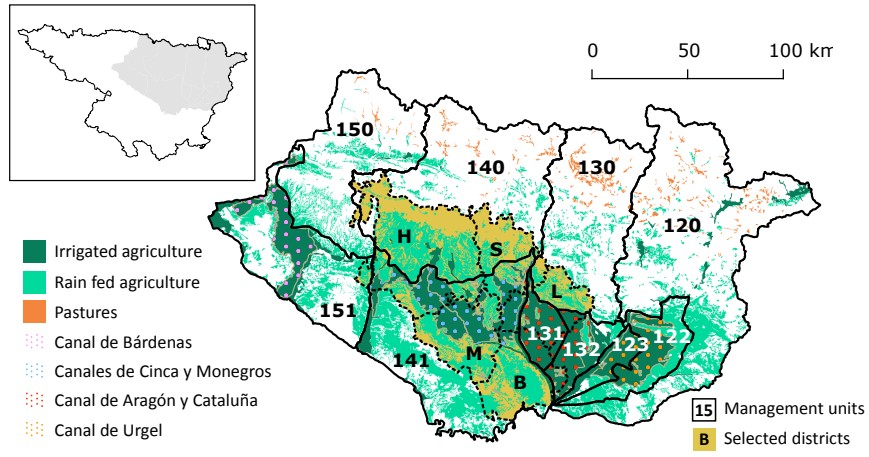

**Figure 1.** The north-eastern part of the Ebro basin with selected agricultural land cover information.

## 2.2 Input data sets

### 2.2.1 Remote sensing data

The analysis focuses on medium resolution global remote sensing products that are related to land surface hydrological and vegetation growth processes. Six commonly used remote sensing parameters were investigated: Precipitation (P), Land Surface Temperature (LST), Normalized Difference Vegetation Index (NDVI), Gross Primary Production (GPP), top soil moisture (SM) and actual EvapoTranspiration (ET). The selected data sets are the following:

**Precipitation (P):** The Climate Hazards Group InfraRed Precipitation with Station data (CHIRPS) is a gridded precipitation data set based on satellite and station data, designed with the main objective to support agricultural drought monitoring. It is a daily, quasi global product, with a resolution of 0.05°. The data set is available from 1981 to near-present. It is based on top cloud temperature measured by geostationary satellites and the Tropical Rainfall Measuring Mission (TRMM) satellite with a rainfall radar aboard. A detailed description of the product can be found in Funk et al. (2015). In this study, monthly aggregated rainfall values have been converted into Standard Precipitation Index (SPI) data sets. The SPI (McKee et al., 1993) is a normalised rainfall anomaly, computed by comparing the accumulated rainfall over a given period with the long-term record. The SPEI R-package developed by Beguería and Vicente-Serrano (2013) was used to calculate SPI for periods of 1, 3, 6, 9 and 12 months. The possibility to calculate the index for different periods is one of the strengths of SPI as it allows to explore the effects of rainfall anomalies of different duration. SPI values calculated for shorter periods are associated to meteorological drought, while those calculated for longer period are often associated to hydrological drought (WMO, 2012).

**Land Surface Temperature (LST):** The MODIS (Moderate Resolution Imaging Spectroradiometer) product MOD11A2 offers day and night LST data sets, available at 1 km resolution as daily and 8-day products (http://modis.gsfc.nasa.gov/data/

dataprod/mod11.php). The daily daytime LST data have been used for the current study. LST is based on longwave emissions in the thermal infrared range (10 to 12 μm).

**Vegetation health (Normalized Difference Vegetation Index, NDVI):** The MODIS Vegetation Indices product (MOD13) provides information on the active leaf chlorophyll, and thus indirectly on the photosynthetically active process. NDVI describes the ratio of the difference and sum of reflected radiances in the red (0.65 μm) and near-infrared parts of the spectrum (0.9 μm). MOD13 is available at different resolutions: 16-day (250 m, 500 m and 1 km) and monthly (1 km) (http://modis.gsfc. nasa.gov/data/dataprod/mod13.php). The monthly 1km data product has been used in the current analysis.

**Gross Primary Production (GPP) and PsnNet:** GPP describes the daily gross carbon flux as a result of the photosynthesic process and is thus suitable to detect the effects of drought on biomass production. The MODIS GPP product (MOD17) applies a light-use efficiency model based on MODIS FPAR (Fraction of Photosynthetically Active Radiation) data, meteorological data and biome-specific parameters. The product also includes Net Photosynthesis (PsnNet), which corresponds to the GPP minus the maintenance respiration for leaves and roots. It is available at 1 km spatial resolution as 8-day composites or annual values (http://modis.gsfc.nasa.gov/data/dataprod/mod17.php) and as monthly aggregates (Numerical Terradynamic Simulation Group, NTSG, http://www.ntsg.umt.edu/). Additional background information can be found in Running and Zhao (2015).

**Soil Moisture (SM):** The soil moisture product considered is taken from the Soil Moisture CCI project, part of the ESA Programme on Global Monitoring of Essential Climate Variables (ECV) (Liu et al., 2011, 2012; Wagner et al., 2012). Three daily products are available (data sets based on active, passive or merged microwave instruments) for the period 1978 to 2014 at a spatial resolution of 0.25 degrees (http://www.esa-soilmoisture-cci.org/). Higher spatial resolution products are only available for certain areas. The product used in the current analysis is the merged CCI SM data set. The data is based on C-band scatterometers and multifrequency radiometers.

**EvapoTranspiration (ET):** There are currently three global data sets of actual ET in the public domain. These are the MODIS ET product (MOD16, https://modis.gsfc.nasa.gov/data/dataprod/mod16.php) developed by the University of Montana (Mu et al., 2007, 2011) and supported by NASA; the Surface Energy Balance System SEBS developed by Su (2002), and the Global Land Evaporation Amsterdam Model (GLEAM) developed by Miralles et al. (2011) which is available through www.gleam.eu. In addition, there are global ET products that are quasi open-access, including the Atmosphere–Land Exchange Inverse Model (ALEXI) being developed by Anderson et al. (1997) from the USDA in conjunction with Hain et al. (2009) from NOAA; the Operational SEBS (SEBSop) of the USGS (Senay et al., 2013; Chen et al., 2016) and the C.M.R.S. Evapotranspiration (CMRSET) published by Guerschman et al. (2009) from CSIRO in Australia. In this paper an ensemble product based on these individual products with accumulated monthly ET values at a pixel resolution of 250 m x 250 m is used. This ensemble ET product (ETens v1.0) is available from the Water Accounting Group of IHE Delft (www.wateraccounting.org). The six individual ET models considered all use different parts of the spectrum, which reinforces the power of this tool.

In order to have one common time interval, precipitation data in mm/day was aggregated by a sum of the daily values for each pixel to obtain monthly data in mm/month and LST and SM data were aggregated by averaging the daily values for each pixel (Table 1). Part of the input data (LST, NDVI, SM, ET, GPP and PsNet) present a seasonal trend. For these, monthly anomalies were obtained by subtracting the mean for the whole period from each monthly average value, using these anomaly

**Table 1.** Selected remote sensing products

| Parameter | Product | Pixel size | Original time interval |
|---|---|---|---|
| P | CHIRPS | 0.05 deg | Daily |
| LST (day) | MOD11A2 | 1 km | Daily |
| NDVI | MOD13A3 | 1 km | Monthly |
| GPP and PsnNet | MOD17 (NTSG) | 1 km | Monthly |
| SM | Merged SM product (CCI project) | 0.25 deg | Daily |
| ET | Ensemble | 250 m | Monthly |

time series as input for the correlation. The remote sensing data has been aggregated per management unit. Pixels with at least 85% of their area within the management unit (30% in the case of soil moisture due to the coarse resolution) were considered in establishing the aggregate value for that unit. For the 1 km resolution LST, NDVI and ET remote sensing products, the results are also analysed by land cover type. This relates to pixels where at least 85% of the area consists of irrigated and rainfed

agriculture land cover classes (see Figure 1).

### 2.2.2  In situ data

In situ data of reservoir levels and inflow and river flow from the basin measurement network were used to calculate the Status Index ($I_e$), a normalised monthly index used by CHE to homogenise the different indicators (CHJ, 2007):

$$\text{If} \quad V_i \geqslant V_{avg} \quad \rightarrow \quad I_e = \frac{1}{2}\left[1 + \frac{V_i - V_{avg}}{V_{max} - V_{min}}\right] \tag{1}$$

$$\text{If} \quad V_i < V_{avg} \quad \rightarrow \quad I_e = \frac{V_i - V_{min}}{2(V_{avg} - V_{min})} \tag{2}$$

where $V_i$ is the value of the indicator for month i and $V_{avg}$, $V_{max}$ and $V_{min}$ are respectively the average, maximum and minimum values of the indicator derived from historical data. Based on this index (which is a value between 0 and 1), the situation under analysis is classified by the authority as normal ($I_e > 0.5$), pre-alert ($0.5 > I_e > 0.3$), alert ($0.3 > I_e > 0.15$) or emergency ($I_e < 0.15$).

The indicators selected by CHE for each of the management areas were used for the analysis presented here. These are the values of reservoir volume for the regulated areas (122, 123, 131, 132, 141, 151), inflow into the corresponding reservoir(s) for the upstream areas (120, 140, 150), and runoff at a selected station for management area 130.

### 2.2.3  Benchmarking data sets

Two different tests were carried out using drought impact data sets as benchmark to assess the ability of remote sensing based

indicators to provide early drought detection information during the period 2000–2012. The short length of the remote sensing data series available was one of the reasons to base the definition of drought we use to build the reference not on a frequency

analysis, in which drought is defined as an extreme event with respect to the historical series, but on the occurrence of drought impacts. The other reason being that what managers need is to identify the conditions that may lead to drought impacts in order to take mitigation actions. In the first test, text-based records of drought occurrence and impacts collected from a review of local news (i.e. qualitative information) were used to reconstruct the onset and evolution of drought conditions during the period of

analysis and as benchmark for the comparison of the remote sensing datasets. Newspaper records were selected as data source because it allowed a systematic collection of impact occurrence data of all affected sectors with a monthly time step for the whole period of analysis. In the second test, the use of crop yield statistics (i.e. quantitative information) is considered as a benchmark of drought impact on agriculture. The correlation of remote sensing data, especially SPI and NDVI, to agriculture yield data has been widely researched and applied (see Bachmair et al., 2016, for a review). This second type of impact data

was included to provide a comparison of the results obtained in the correlation to text based impact data and results obtained with the most commonly used type of impact data and discuss the advantages and limitations of one with respect to the other.

Text-based data sets were collected from a review of regional news. "El periódico de Aragón", the second largest newspaper in average daily circulation in the Aragón region was selected for the review because it has an online record going back to September 2001. All news items containing the word "drought" were reviewed and relevant records of drought events

and impacts referring to the area of study were tabulated. For each entry the location, period, description, and, in case of reported impacts, the affected sector were noted. The affected sectors were labelled as *rainfed agriculture*, *irrigated agriculture*, *livestock*, *water quality*, *fire*, *water supply*, *energy* and *others*. The records of drought occurrence are classified according to the source of the information, making a distinction between non-official sources such as journalists and water users, labelled *mention of drought occurrence* in Figure 2, and official sources labelled as *drought acknowledged by the authorities*, *ongoing*

*mitigation measures* and *periods retrospectively defined as anomalously dry*. This last type corresponds mainly to news about the publication or communication of analysis performed by the scientific community or the water mangers describing an ongoing or past drought.

The limit between indicators and impacts is not always clear. For example, low flow or reservoir levels are considered an impact of meteorological drought in some analyses, while these serve as indicators of hydrological drought in others. Here we

limit the definition of drought impacts as the effects of drought on people, economy and/or the environment.

Crop yield data of winter cereals both for irrigated and rainfed cropping systems was obtained for the five selected districts in Huesca (H, S, L, M and B). Winter cereals are the cereal crops that are planted in the autumn and they are the crops that cover the largest surface area. Their importance for the region results in better data availability than other crops and for this reason this type of crops were selected for the analysis. Only winter cereal crops with larger cultivated areas were considered:

two and six-row barley (irrigated and rainfed), wheat (irrigated and rainfed), and rice (irrigated). The two and six-row barley types refer to the number of fertile spikelets in the spike.

## 2.3  Correlation between remote sensing data and the benchmarking data sets

The correlation between each of the remote sensing parameters and both the timeline that aggregates all types of drought events records and the timeline that aggregates all types of drought impacts (Figure 2) was analysed in terms of strength of the

relationship and anticipation. The strength of the relationship is a function of the predictability of the occurrence of drought and drought impacts provided by the remote sensing time series. Anticipation reflects the ability of the remote sensing data sets in providing early information and gaining time to undertake actions. The aim of this analysis is to identify the datasets that can be useful for operational drought detection at the basin scale. Drought detection in this case is closely related to the predictability of impacts, as the conditions that need to be detected are those that may lead to impacts. However, these impacts do not necessarily occur immediately; their occurrence can be delayed as the effects of drought propagate through the different components of the hydrological cycle. To identify the remote sensing parameters that represent conditions that anticipate the occurrence of drought impacts, and therefore have potential to support the prediction of drought, we explore the correlation between the remote sensing data and the drought events and impacts at different time lags. The benchmark datasets were compared to the variables represented by the remote sensing time series in the 24 preceding and following months. While using correlation in this way may say less about the long term correlation of two time series, it does provide insight in the relationship between correlation and lag.

The sample cross correlation function (CCF), $r_{x,y}$, was used for the analysis. The CCF can be expressed as (Chatfield, 2004):

$$c_{x,y}(\tau) = cov(X_t, Y_{t+\tau}) \tag{3}$$

$$r_{x,y}(\tau) = \frac{c_{x,y}(\tau)}{\sigma_x \sigma_y} \tag{4}$$

For $\tau = \pm 1, \pm 2, \ldots$, where $\tau$ is the lag and $\sigma_x$ and $\sigma_y$ are the standard deviations of the time series $x_t$ and $y_t$. The set of $c_{x,y}$ coefficients corresponds to the cross covariance function. The CCF function as implemented in R (R Core Team, 2016) was used for the calculations. To detect possible issues related to the stationarity or ergodicity of the series, their time autocorrelation and partial autocorrelation plots were considered.

The reference drought periods used for the correlation provide a binary record; indicating the occurrence or non-occurrence of drought events in each month, without quantifying their intensity. To obtain an insight into the severity of the events, the use of annual crop yield data was explored. The correlation of each annual crop yield value to the monthly values of the remote sensing time series from the start of the hydrological year in September to the end of the following calendar year was analysed. This was done to detect the key months in which the occurrence of drought conditions lead to impacts on the (annual) crop yield. The comparison was performed for three rainfed areas and three irrigated areas. These were selected to correspond with the management units so that a relation could be established with the areas of influence of the reservoirs (Figure 1). The areas selected included the rainfed agriculture areas of Hoya Huesca (HH0, corresponding to management unit 140), Monegros-Bajo Cinca (MB0, corresponding to management unit 141), and in the five districts together (AA0) and the irrigated agriculture in Hoya Huesca-Monegros (HM1, corresponding to management unit 141), La Litera-Bajo Cinca (LB1, corresponding to management unit 131), and in the five districts together (AA1).

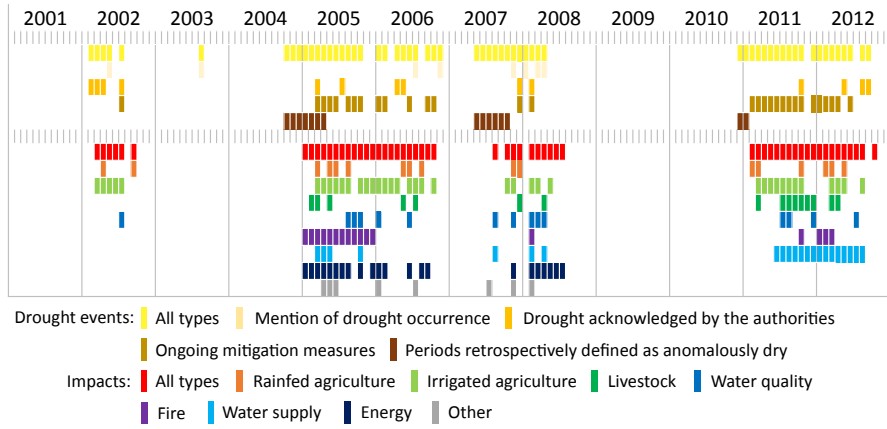

**Figure 2.** Timeline of drought events (upper part) and impacts (lower part) for the north-eastern Ebro basin during the period 2001 to 2012 based on a comprehensive local newspaper review.

## 3 Results

### 3.1 Drought events and impacts

The timelines of drought events and impacts derived from the review of local news are illustrated in Figure 2. Three drought events can be distinguished in the Ebro basin: a short drought event at the beginning of 2002, a multi-year drought from the end of 2004 to the spring of 2008 and a shorter duration drought during 2011 and 2012.

The first colored row (yellow) in the figure represents the months in which drought was taking place according to the records found in the newspapers. The first line of the second block (red) reflects the occurrence of drought impacts described in the newspaper, while in the following rows these impacts are disaggregated by the affected sector.

Based on the records gathered from the newspaper records, the following descriptions of the hydrological years affected by drought episodes were constructed:

In 2002, after a dry winter the availability of water in the reservoirs was low. A first reference to drought in the press appeared in February 2002. At the start of the spring, which is the beginning of the irrigation season, water curtailments were reported for Bardenas irrigation system. In the beginning of April, Agricultural Associations reported losses of 20% of rainfed cereal crops in Aragon and at the end of the month the impact of drought in the area was acknowledged by the Ministry as well as by the local government. In July, the flow of the Ebro in Zaragoza was half of the minimum 30 m$^3$/s that has been set to warrant water quality. In September a reduction of 40–70% in olive oil production in the areas of Bajo Cinca, Medio Cinca and La Litera was reported in the news. General mention of impacts on pastures, hydroelectricity production and employment in the primary sector during that drought period appeared in retrospect, but these reports did not go into detail.

The hydrological year 2004–2005 was depicted as the driest on record. The combination of cold and dry conditions during the first part of 2005 produced significant losses in the agriculture and livestock sectors. First impacts were reported in February

2005 (lack of pastures production after five months without rain). From then, and until September 2006, the newspaper reflected a succession of impacts in different sectors, including all crop types, pastures, forests, livestock production, water supply to the population, wildlife, economy, recreational activities, hydroelectricity, water quality, employment and politics. The drought was already acknowledged by the authorities since March 2005, and the first mitigation measures were announced shortly after.

This was that the regional government increased to 50% the area of land, rainfed or irrigated, that could be set aside to remain fallow. In June, aid measures were approved by royal decree.

Reservoir levels increased during the first half of the hydrological year 2005–2006, but the system failed to recover completely from drought before levels started decreasing again in April 2006, and at the beginning of the summer levels were lower than the previous year. After a hot summer, storage started to recover again and in December 2006 the government considered

the drought as having ended. Intense rains starting in February 2007 were followed by a period of precipitation deficit from May to February 2008. A few problems of water supply to certain villages were reported in August 2007 and flows were below the minimum required to warrant water quality in October. Impacts on agriculture and hydroelectricity started to be reported again in October. Abundant rains during spring 2008 constituted a first step towards the end of the drought episode.

The hydrological year 2010–2011 was characterised by lower than average precipitation and high temperatures. In February

2011 the newspaper showed the first reference to an emerging drought and its impact on the sprouting of winter cereal. This drought especially affected the Bardenas irrigation district. The Riegos del Alto Aragón and Canal de Aragón y Cataluña districts were also affected. All the systems managed to reach the end of the irrigation season, but with restrictions of more than 60% on water quotas. Grapes and olives were the most damaged crops, but in general the food production in the area was defined as satisfactory at the end of the season. The following hydrological year (2011–2012) started with low reserves and a

dry winter and spring, with the exception of November, which was a particularly wet month. In particular, the middle sector of Huesca revealed drought affected areas. Extensive livestock farming, fodder and cereal production were the most impacted sectors. The risk of fire was reported to be high, even during the winter, which translated in a higher number of fires.

## 3.2   Correlation of text-based records and remote sensing indicators

The information on drought occurrence and impacts obtained in the previous step was used as a benchmark data set to assess

the ability of the remote sensing based data sets to provide early detection. Figures 3 and 4 present the results of the cross correlation of the remote sensing data sets to the timelines of drought events (i.e. upper records in Figure 2) and impacts (i.e. all other records in Figure 2) respectively. The central line ($x = 0$) corresponds to the correlation of the two data sets in the same month. Negative values of x refer to correlations between impact time series at time t and remote sensing values at each of the 24 months before t ($\tau = -1$, $\tau = -2$, ..., $\tau = -24$). Strong correlations on the left side of the central line reflect the

ability of the data set to anticipate the occurrence of drought events and impacts. The positive side of the plots reflects the correlation of the drought occurrence and impact series with the values of the different datasets in later months. This type of correlation appears if the conditions that define the start of the event or impact occurrence last longer than one month. The positive correlations of the timelines of drought occurrence and impacts with the values of the indicator datasets with lags over

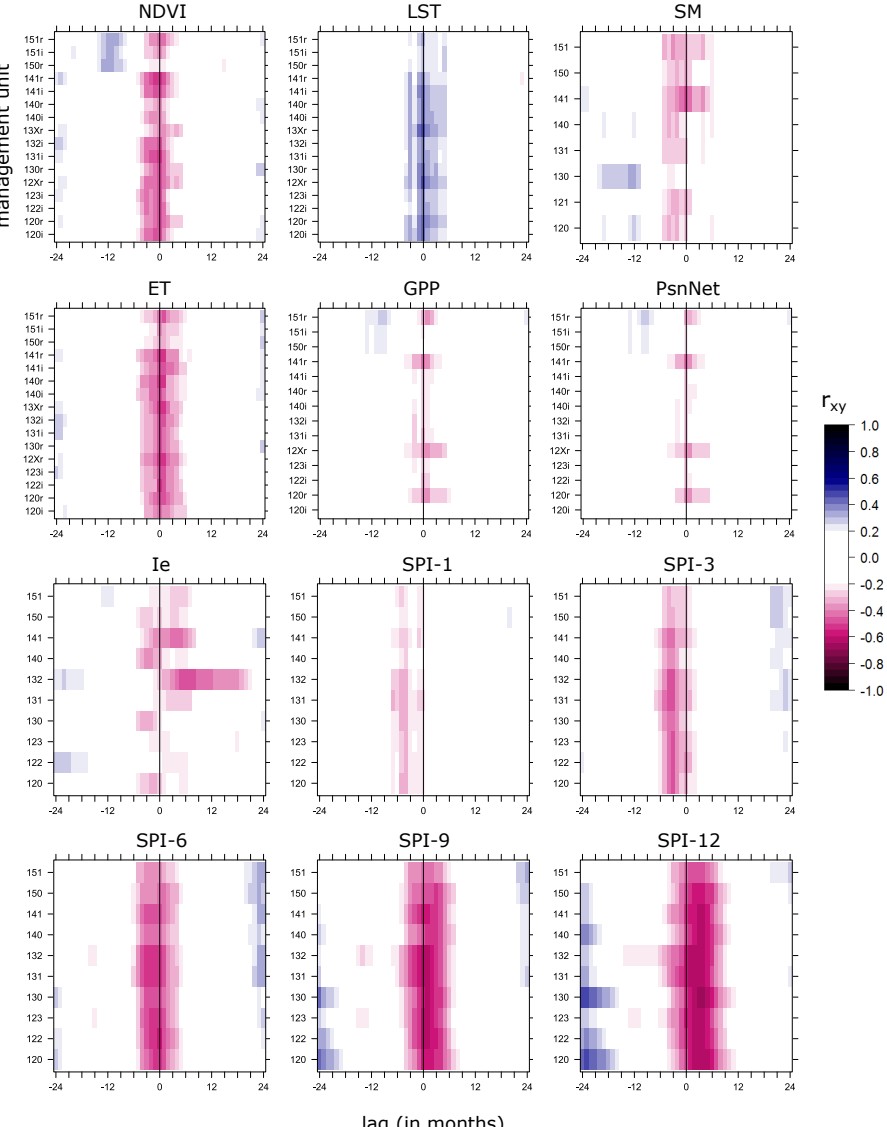

**Figure 3.** Cross correlation of drought indicators and drought events at multiple time lags. The numbers in the y-axis represent the management areas depicted in Figure 1. For NDVI and LST, irrigated (i) and rainfed (r) crops within the areas are distinguished. The x-axis represents the shift in months between the two data sets. The indicator built from in situ data ($I_e$) is also included.

one year (most notably at -15 and ±24 lags) are casual correlations. Since the analysis presented here focuses on anticipations within a period of one hydrological year, the correlations should not be affected by this issue.

Figures 3 and 4 have similar correlation patterns, with the second showing higher anticipation. This result was expected because Figure 3 is based on records reflecting climatic and hydrologic anomalies and deficits, and these processes precede the impacts. SPI shows the strongest correlations for both events and impacts. For SPI values calculated for longer aggregation

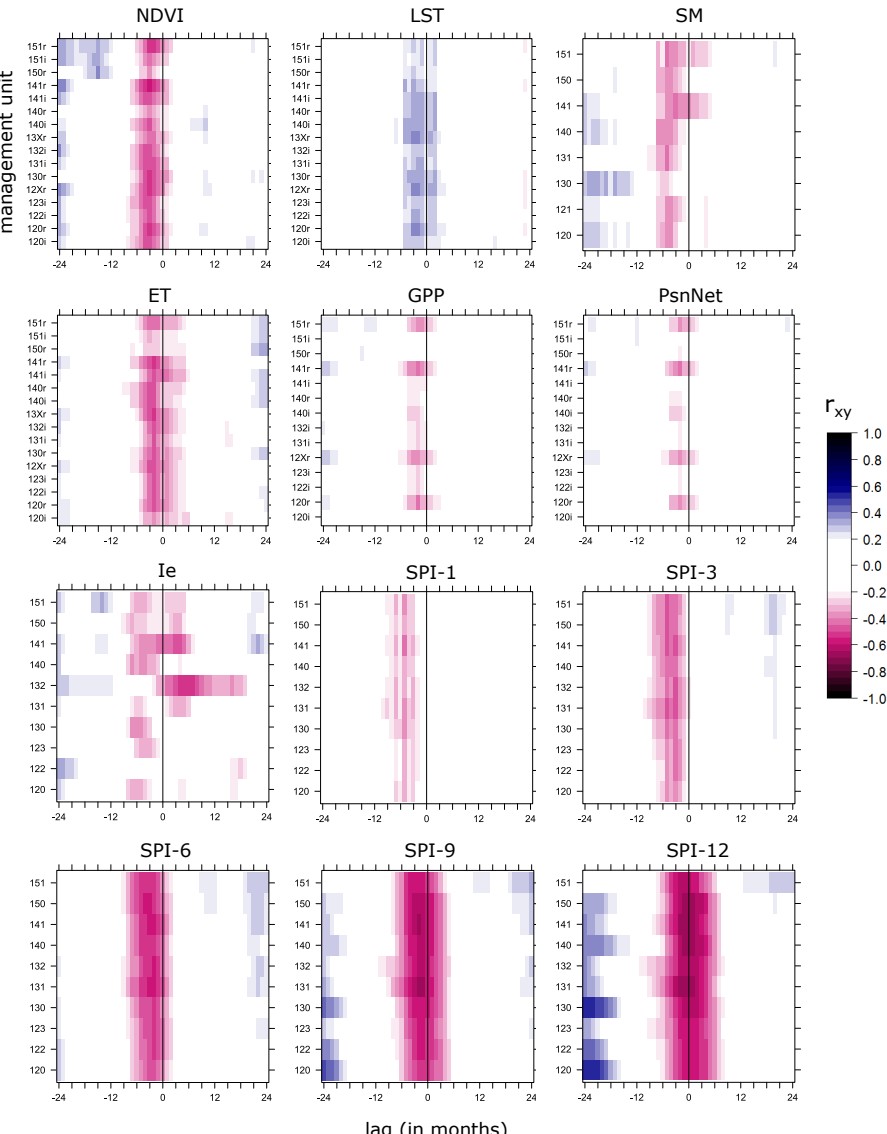

**Figure 4.** Cross correlation of remote sensing data sets against the timeline of reported drought impacts (all types).

periods the correlation grows stronger, while the anticipation is slightly reduced. The best correlation-anticipation relationship is obtained for SPI-6 and for SPI-9. For these indicators, the correlation is also stronger in the southern areas. This is probably because most of the socio-economic activities are concentrated in these managed areas, and therefore the impacts and media attention are likely to be higher. The results show that SPI-6 and SPI-9 are most suitable for predicting impacts, together with NDVI and ET; achieving an anticipation of 6 months with a sufficient correlation ($r^2 > -0.6$). This provides useful information for activating drought mitigation measures. Soil moisture also shows good anticipation, albeit with weaker correlations. NDVI

and ET data sets show a strong negative correlation with drought occurrence and impacts, which would be expected from a biophysical perspective. NDVI shows better anticipation, preceding the impacts in most of the units by more than 6 months. ET shows a slightly stronger correlation in the rainfed areas, while no distinction is seen between rainfed and irrigated areas for LST and NDVI. LST has a positive correlation because evaporative cooling is diminished during drought events which prompts the land surface temperature to rise. LST correlation to events is stronger than to impacts, but the degree of anticipation is lower for the former. Indices derived from both GPP and PsnNet present weak or no correlation for most of the areas, with only some of the rainfed areas showing moderate positive correlations. In situ indicators show varied level of anticipation for the different areas. Most of them provide early information on drought occurrence and impacts from 6 to 9 months in advance, but there are two areas where the indicator offers no anticipation (management unit 140) or even no correlation with the benchmark data sets (management unit 123).

The time plots obtained for each of the parameters present no trends or discontinuities and the values in the autocorrelation plots show that the autocorrelation diminishes quickly with increasing lag. An exception are the series of the reservoir indices. In that case, for some of the series it is not clear from the plot if the series is stationary. For one of them (management unit 122) it clearly is not. This management unit corresponds to a reservoir (Rialb) that started to be filled in the year 2000 and therefore the levels cannot be considered stationary for the period of study. Most of the autocorrelation plots for the reservoir level series present a small peak of autocorrelation at a lag of 12 months, and one of them (management unit 132) presents autocorrelation values declining more slowly (significant values until lag 20). In the $I_e$ plots in Figures 3 and 4 it can be clearly seen that the two management units that do not satisfy the conditions for stationarity (management units 122 and 132) are those (at least two out of the three) that do not present anticipation. For the remaining products, autocorrelation for ET, LST, GPP, PsnNet, SM and SPI-3 dissipates mostly at a lag of 2 months. For SPI-1 it goes quicker and is non-existent in some cases. NDVI takes 3-4 months and for SPIs with longer accumulation periods (SPI-6, 9 and 12) the correlation dissipates slower (4, 6 and 8 months respectively), which is inherent to the product.

### 3.3  Correlation of crop yield and remote sensing indicators

The results of the correlation analysis between the remote sensing data time series and the annual crop yield for the main irrigated and rainfed cereal crop types in the selected districts in Huesca are represented in Figures 5 and 6. Every parameter is tested for the six areas. The crop types are represented on the y-axis and the months on the x-axis. The latter spans from the start of the agricultural year in September to the end of the following calendar year. The colour gradient reflects the sign and strength of the correlation, while the size of the inner grey circle corresponds to the reliability of the correlation.

NDVI and ET present some of the strongest positive correlations, especially between the remote sensing measurement during the spring (MAM) and the yield of rainfed crops. LST shows also strong correlations with rainfed crops in March and at the beginning of the season in September (S). The pattern is less clear for irrigated crops, probably because their water supply is less dependent of the rainfall. The strongest correlations in this case appear for rice crops with ET and NDVI, mainly at the start of the year.

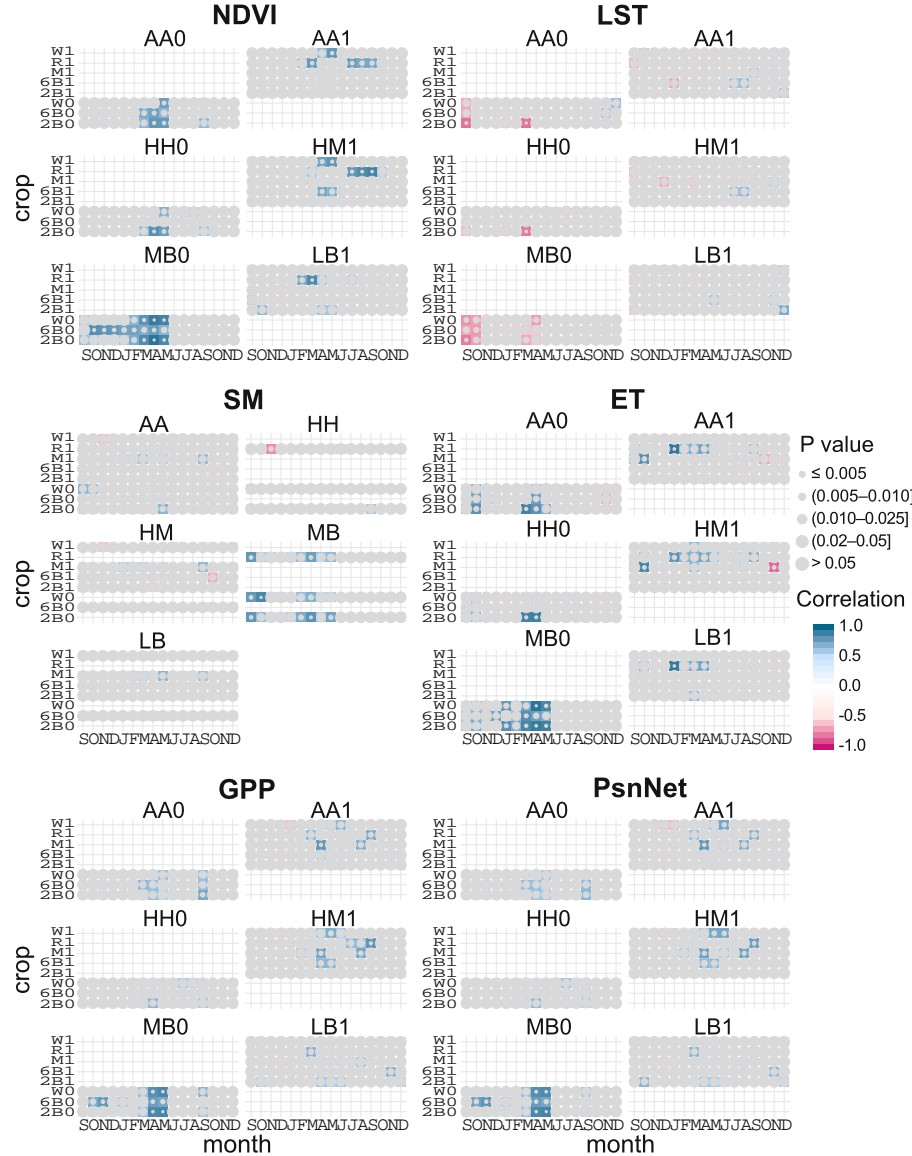

**Figure 5.** Correlation between remote sensing drought indicators and crop yield data. Rainfed areas and crops are marked with 0 and irrigated areas and crops with 1. The crops are irrigated and rainfed wheat (W1 and W0), irrigated rice (R1), irrigated maize (M1), irrigated and rainfed six-row barley (6B1 and 6B0), and irrigated and rainfed two-row barley (2B1 and 2B0).

Despite irrigated crops directly depending on reservoir supply, only rice shows significant positive correlations with the index based on reservoir levels for the two irrigated areas tested (HM, corresponding to management unit 141 and LB corresponding to management unit 131). The reason can be that rice is especially drought sensitive, since it has shallow roots and consequently a low depth of readily available soil water, which is the fraction of total available soil water that crops can obtain from the root

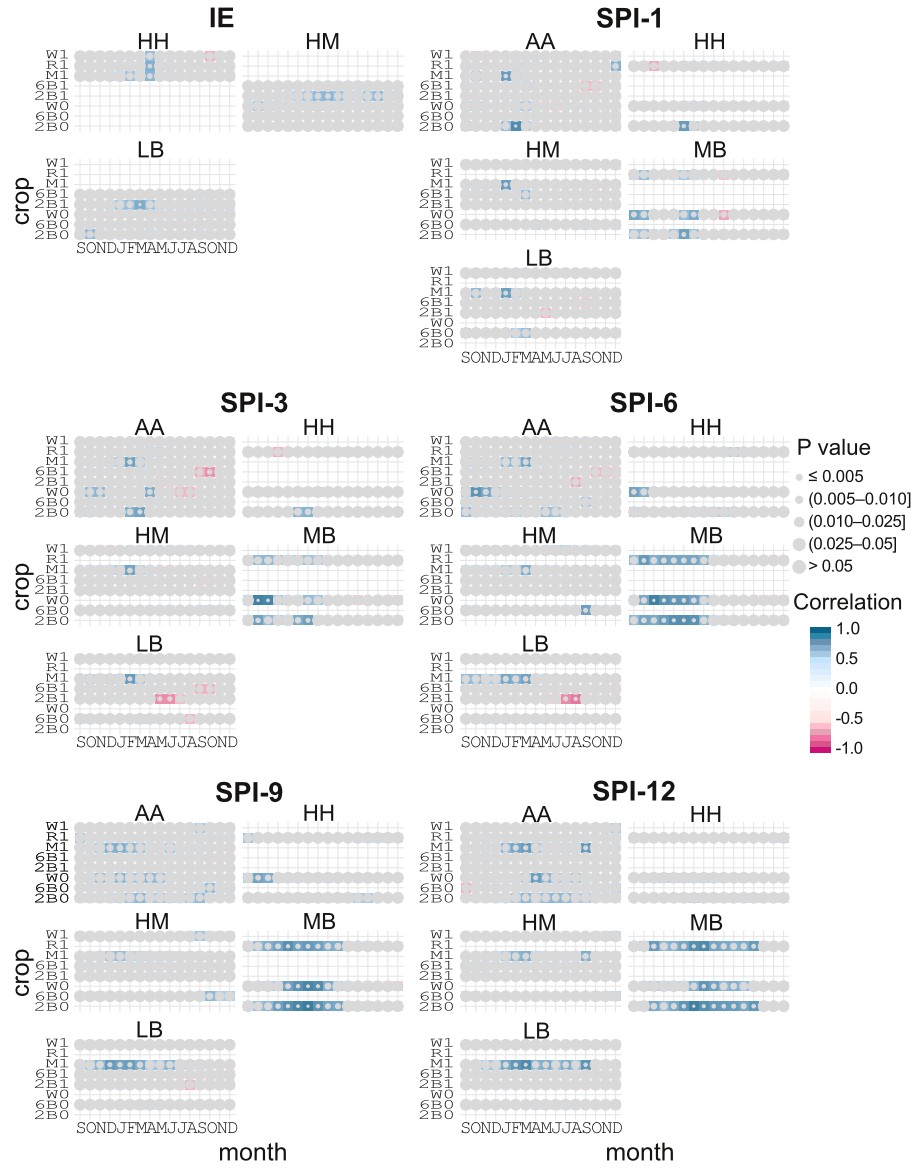

**Figure 6.** Correlation between drought indices (SPI and Reservoir index) and crop yield. Rainfed areas and crops are marked with 0 and irrigated areas and crops with 1. The crops are irrigated and rainfed wheat (W1 and W0), irrigated rice (R1), irrigated maize (M1), irrigated and rainfed six-row barley (6B1 and 6B0), and irrigated and rainfed two-row barley (2B1 and 2B0).

zone without experiencing water stress. This fraction is 0.2 for rice (Allen et al., 1998) and higher for the rest of the tested crops, that therefore experience stress when more moisture is depleted. The rainfed crops in the HH area (corresponding to management unit 140) show correlation with the Status Index based on reservoir inflow, in April. Soil moisture, GPP and PsnNet do not show a clear pattern against reported crop yield, except for a strong positive correlation in one of the areas

(MB). These correlations appear in the spring (especially for GPP and PsnNet) and at the beginning of the hydrological year. SPI has positive correlations at the start of the season, which are particularly strong in the MB area. In some cases also a negative correlation during the summer emerges, especially for shorter term SPIs. It can also be observed that the stronger correlations appear later with longer term SPIs.

Rainfed two-row barley (2B0) in March stands out as the crop with the stronger overall correlation with the different indicators. 2B0 is one of the major crops in the area, with a maximum cultivated surface for the period 2000–2012 of 170,914 ha of the total of 204,614 dedicated to herbaceous crops (in 2008), and a minimum cultivated surface of 130,764 ha (2012). Maize is the second commonest crop with an average crop surface of 41,292 ha during the period. Figure 7 illustrates the correlation of two-row barley to each of the indicators for the rainfed crops in the Monegros-Bajo Cinca area (MB0), which has been selected

as an example. This shows the crop yield for the different years against the value of each respective indicator.

     Three years stand out in Figure 7 for having extreme low indicator values (high in the case of LST) for all variables: 2005, 2008 and 2012. Values are also low for year 2000 for the data sets for which it is available. These three years correspond with hydrological years of reported impact in the area identified in the previous section. The lowest crop yields were obtained in 2012 (1342.5 kg/ha). Accordingly the remote sensing parameters present some of the lowest (highest in the case of LST)

values for the period. Only SPI-6 presents a value that is well above the minimum. This is caused by November 2011 being a particularly wet month in the middle of the drought period, thus moderating the value of SPI-6. For longer term SPI values this positive anomaly is compensated by the negative anomalies of the rest of the months.

     Crop yields were very similar in 2005 and 2008 (1662.3 and 1800.1 kg/ha respectively) and so is the behaviour of most of the variables. The main differences appear in LST, with the 2005 LST for March being more than 2 degrees higher than in 2008

(23.8 and 21.3 degrees respectively), and SPI-3, which is less extreme in 2008 (–0.68 compared to –1.51 in 2005). The reason for this difference is the earlier start of spring rains in 2008. Both hydrological years start with an exceptionally dry period that extends to April in 2005 and to March in 2008 after which the spring rains improve the situation.

     There is a second group in the middle sector of the plots that includes the rest of the years for which drought impacts on rainfed agriculture were reported in the analysed media, 2011 (3551 kg/ha), 2006 (3857 kg/ha) and 2002 (4249 kg/ha),

together with the hydrological year 2006–2007 (3115 kg/ha), for which no impact was reported in the regional press. March values of SPI-3 for these years are close to the mean and only 2002 presents strong negative anomalies for SPI-6. In 2006 and 2007 the precipitation deficits start in April and May respectively and for 2006 the impacts are reported only after that month. Hydrological year 2001-2002 shows dryer autumn-winter conditions according to SPI-6.

     The results of this second tests present a consistent behaviour of the indicators with respect to the different levels of crop yield

among the analysed years in rainfed areas. As in the previous test, NDVI, ET and SPI stand out for having stronger correlations. Most indicators present similar March values for the years of severe drought, clearly differentiated from the behaviour of years of moderate drought and years of no drought. The only exception is LST, in which a year where drought was not reported and yields were normal, such as 2009, has similar LST values in March than the years of severe drought. This indicates that LST may not be a good indicator of drought on its own, but can still be useful in combination with other indicators.

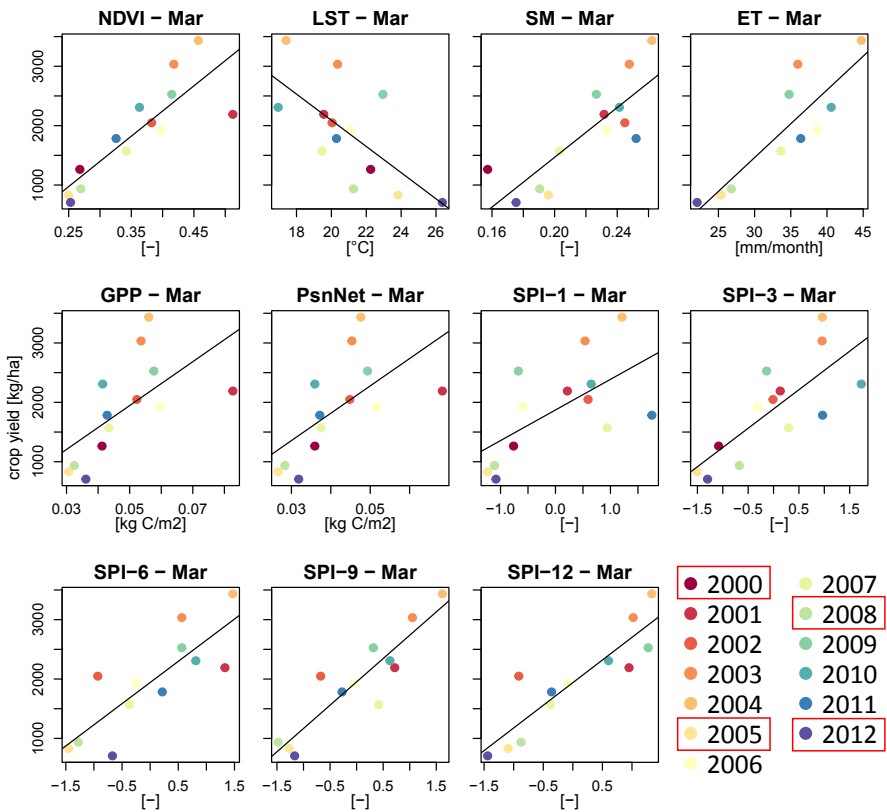

**Figure 7.** Correlation of the remote sensing drought indicators for the month of March to annual rainfed two-row barley yield in Monegros-Bajo Cinca districts (MB). For LST, NDVI, SM, ET, GPP and PsnNet monthly means were used.

## 4 Discussion

### 4.1 Use of impact records as drought reference

The review of text-based records allowed a detailed reconstruction of the drought events during the period studied. The cross correlation of the timelines of drought events derived from this review to the indices derived from remote sensing data revealed the potential of the latter to provide early detection of drought events. However, this binary information has the limitation that it does not allow to objectively quantify the severity of the events. For example, in the case of rainfed agriculture, the information on impacts collected from the newspaper does not allow for differentiation between those years in which production was extremely low as a consequence of drought conditions and those years in which production was only partially affected by drought. Other studies have suggested a link between impact severity and the number of records reporting them (e.g. Hernández Varela et al., 2003; Bachmair et al., 2016), but this needs to be taken carefully since the media coverage of a drought event is highly influenced by the socio-political context in the affected area (Sonnett et al., 2006; Llasat et al., 2009).

A few additional aspects concerning reliability were noticed while processing the records from the press: (i) Accuracy: The information on drought occurrence reported in the newspaper may not be accurate. For example impacts due to other causes may be attributed to drought, or other phenomena such as normal summer shortages may be described as drought. This issue was the reason to classify the records of drought occurrence according to the source of the information to make a distinction between official sources such as mandated authorities, managers and scientists; and non-official sources such as journalists or water users. This second type of source is the one that is most susceptible to accuracy issues. Particularly for the case of the mandated authorities there are clear procedures with which drought is officially acknowledged, which are defined in the drought management plan. In the records reviewed, only the mention of drought conditions recorded in 2003 is not backed-up by mention of drought from official sources during the same period, and may therefore be regarded as a misuse of the word. Thus, we consider accuracy issues to have little impact on results. (ii) Completeness: Reporting of drought occurrence in the newspaper is not systematic and therefore some impacts may be missing. In Figure 2 some unlikely situations can be identified. For example impacts to livestock in May and July 2006, but not in June. Records referring to specific types of impacts are more likely to have gaps. However, when all types are aggregated, part of the gaps in each of the disaggregated data sets will likely be filled with records from the other data sets. (iii) Scale: Drought events affecting only a small area within the region covered by the newspaper may not be reported. The results of the test with crop yield data show values for the hydrological year 2006-2007 for which no drought impacts were identified in the reviewed regional newspaper that are similar to three other hydrological years for which drought impacts were recorded. Local press for the specific area of the test (Alto Aragón), however, reported a lack of rain from October to March, aggravated with high temperatures, in Monegros and Bajo Cinca that had an impact on rainfed cereals and pastures. (iv) Bias: Public or political interest or concern about drought (or even scarcity of other relevant news) can motivate overstatement of drought impacts. These do not have an influence in our analysis since we are only considering binary data of occurrence or non-occurrence, but this issue could have a significant impact on the reliability if the records were used to estimate the severity of the event.

The length of the period of analysis does not have an influence in the identification of drought events based on impact records. However, having a longer series, and therefore potentially a larger number of drought events, would provide more robust results in the correlation analysis. Ideally the results should be updated as the period of record of remote sensing data grows.

The drought events identified by the textual search for a sector of the Ebro basin correspond with events observed at a larger scale. For example, Spinoni et al. (2015), use an indicator that combines three precipitation and potential evaporation-based indices to identify the drought events occurred in different regions of Europe during the period 1950-2012. Following that approach they identify three drought events for the Iberian Peninsula for the period 2000-2012 that match the ones obtained by the textual search, with the difference that the event starting in the hydrological year 2004-2005 has a shorter duration. This is caused by the different spatial scale of the analysis. While most of the basins in Spain received normal precipitation during the hydrological year 2006-2007, in the Ebro basin, and especially in the inner part of Catalonia, it was still low during that year (MMA, 2007).

Crop yield data, on the other hand, allowed for a more objective identification of the drought events that had higher impact on agriculture, though the yield data does have the disadvantage that may only be reported on an annual basis. March was the month that presented higher correlations. This is in agreement with the results obtained by Vicente-Serrano et al. (2006) who observed a higher correlation between barley crop yield and NDVI for the month of March at a location in the Ebro River valley. The examination of the behaviour of the remote sensing parameters in the years with similar yield values provided an insight on the reliability of the parameter as an index. Similar values of the parameter for years of similar final crop yield indicate the robustness of the indicator. For the period of analysis, the occurrence of drought in the years in which low yields were obtained is confirmed by the media records, but despite that the water availability is a determinant factor for rainfed winter cereal yield, other factors such as frost, floods, plagues and diseases could have further reduced the annual yield. However, for March values, there are no anomalies that suggest these factors had a strong influence in the low annual yield values. The lowest crop yields were obtained in 2012. This is in line with the information in newspapers reporting that drought that year especially affected cereal production in the middle sector of Huesca, the area of focus of this test.

Crop yield data can also be a useful reference to identify thresholds of drought severity classes. These thresholds could be derived based on the differences observed between the groups of years with severe, moderate and no drought conditions, although a longer data series than was used in this study is recommended to provide a more robust estimate of threshold values.

There are several factors that play a role in the severity of the impacts due to drought conditions, including coping capacities and water management (e.g. drought may not lead to impacts in irrigated areas). Variations in these factors can alter the relationship between the indicators and the impact. It should also be noted when using drought impacts as a benchmark of drought occurrence, that the absence of certain types of impacts as a result of sound drought management does not imply that there is no drought (Smakhtin and Schipper, 2008), though even with perfect management there will always be some kind of impact. For example, a reduction of income as a consequence of substituting the usual crops by less productive alternatives with lower water requirements constitutes a clear impact, even if the yield in kg/ha is not affected. The influence of management is probably also the reason for irrigated land showing less clear correlation patterns than drought in rainfed areas in both analyses. A wider view that considers as many different types of impacts and affected sectors as possible can help overcome the effect of management when using this type of data as a benchmark of drought occurrence. Initiatives such as the US Drought Impact Reporter and The European Drought Impact report Inventory can play a useful role in providing that broader view.

## 4.2 Early drought detection with remote sensing products

Early information on emerging droughts benefits mitigation strategies by increasing the time available for managers and affected communities to take action. The requirements for drought early warning range from a few weeks to several months (UNISDR, 2009). The results show the potential of the tested products to anticipate up to some 6 months reported drought impacts at basin scale. SPI, NDVI and ET products stood out in both analyses as particularly suitable datasets to detect early stages of drought at the basin scale and anticipate drought impacts. However, while for most products the autocorrelation dissipates at a lag of 2 months, for NDVI and SPI-6 it takes 3-4 months and this can have an influence in NDVI and SPI-6 showing stronger correlations. For SPIs with longer accumulation periods (SPI-9 and 12) the correlation dissipates even slower.

The weaker correlations obtained for SM data in the first test may be due to the coarser spatial resolution of the data set. Higher resolution soil moisture products (e.g. Scott et al., 2003; Alexandridis et al., 2016) could be considered for future studies. The reason for the weak or no correlations between both GPP and PsnNet and the text-based records may lie in the formulation of the MOD17 product. Indeed, limitations of the product in capturing spatial and temporal variability in croplands have been reported (Verma et al., 2014; Zhang et al., 2012).

The trade-off between the anticipation of the information and its reliability is also illustrated by the results. The lower reliability associated with earlier information detection of conditions that may lead to drought implies that often the situation may not evolve into a drought event. However, that information is still highly valuable as it allows the stakeholders to get ready to undertake mitigation actions if necessary.

The remote sensing products tested can enhance early warning capacity and therefore contribute to the shift from reactive to proactive management recommended by the European Commission (Commission of the European Communities, 2007) and the United Nations (UNISDR, 2009) and being undertaken by many institutions (Iglesias et al., 2009). As remote sensing data products generally have a global coverage, this contribution would therefore be especially useful in areas with less in-situ data available. Yet the most informative indicators of drought occurrence may vary depending on specific characteristics of the country or basin, such as management practices or dominant water uses (Stagge et al., 2015). Remote sensing products also have the potential to provide information at a finer spatial detail than the management units and land cover classes considered in this study, allowing the detection of local drought events that may remain unnoticed when the pixels are aggregated to the scale of the land cover classes considered.

## 5   Conclusions

The aim of this research was to test the ability of remotely sensed datasets to detect early stages of drought at the river basin scale, with particular attention to their capacity to anticipate drought impacts and gain time to inform operational land and water management. Media records from regional newspaper proved to be a helpful source of information that allowed a detailed reconstruction of drought events and impacts. The analysis using this data as a benchmark revealed the potential of the tested medium resolution remote sensing products to anticipate reported drought impacts on irrigated and rainfed areas at basin scale up to some 6 months. The best correlation-anticipation relationships were obtained for SPI, NDVI and ET. SM and LST also showed potential to anticipate drought, but with weaker correlations. GPP and PsnNet from MOD17 presented weak or no correlation for most of the areas, with only some of the rainfed areas having moderate positive correlations. The index based on in-situ data currently used in the basin also provides early detection, and with the exception of two of the management units, the anticipation of drought impacts is better than that provided by the remote sensing indicators. However, the correlation of the indices based on SPI, NDVI and ET to anticipate drought impacts was found to be stronger. The use of quantitative impact data of crop yields as benchmark showed a consistent behaviour of the remote sensing indicators with respect to the different levels of crop yield in rainfed areas among the analysed years. SPI, NDVI and ET stand out for having stronger correlations,

reinforcing the findings of the first analysis. In both analyses drought on irrigated land showed less clear correlation patterns than drought in rainfed areas.

Altogether, the results confirm remote sensing products' ability to anticipate reported drought impacts and therefore provide a useful source of information to support drought management decisions at the basin scale. However, further analysis of
5 manager's information requirements and response options is required to better assess the usefulness of these type of products in informing specific operational drought management decisions.

## 6 Data availability

The remote sensing data used in this research is openly available. The sources are mentioned in section 2.2.1. In situ data from the basin measurement network can be downloaded from the Ebro basin authority site (www.chebro.es). Crop yield
10 data can be downloaded from the site of Aragon government (www.aragon.es). The data set derived from the review of newspaper records is made available as supplementary material. The corresponding news articles can be accessed online at www.elperiodicodearagon.com.

*Competing interests.* The authors declare that they have no conflict of interest.

*Acknowledgements.* This research received funding from the European Union Seventh Framework Programme (FP7/2007-2013) under grant
15 agreement no. 603608, "Global Earth Observation for integrated water resource assessment": eartH2Observe. This work is a contribution to the Hymex Drought and Water Resources Science Team. The authors would like to thank the reviewers for their thoughtful comments.

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
