# Peer review of "The predictability of reported drought events and impacts in the Ebro Basin using six different remote sensing data sets"

_Hydrology and Earth System Sciences, 2016_

## Referee Comment (RC1) · A. Wright (Referee) · 12 Feb 2017

A. Wright (Referee)

a.howells@pgr.reading.ac.uk

General comments:

The paper is very well written. The sentences are clear and the graphs are well presented. It is an interesting piece of research, with a good description of the paper elements. The paper can be beneficial for drought management and early warning. It explores the performance of some indicators in early drought identification. The paper does not derive drought threshold values but assesses the potential of remote sensing datasets for drought threshold derivation. The cross correlation function used in this research for drought anticipation, assumes stationary and ergodic data. It is not clear from the paper how the data was tested for the aforementioned criteria.

[Figure]

Specific comment:

1- You have used the newspaper data as benchmark data for drought occurrence. Could you please elaborate on why you decided to choose newspaper data? How reliable do you think the newspaper data is? What is the false alarm rate?

2- For the daily remote sensing data, you have used "monthly aggregate". Could you please explain more on how the aggregate was obtained?

3- You have used the cross-correlation function as expressed by (Chatfield, 2004), which assumes the bivariate data is stationary and ergodic (on page 122 of Chatfiled, 2004). How did you test if your data was stationary and ergodic?

4- You have used the cross-correlation function as a method of drought anticipation. Can the same technique be applied for foreseeing the end of droughts?

5- As you explained in figure 3 and 4, the values on the left side of (negative side) of the central line, show how early the remote sensing data have anticipated the newspaper headlines on droughts, or anticipated the decline in crop yield. Could you please explain what the values on the right side (positive side) of the central line show?

6- Following from the previous comment, regarding figures 3 and 4, could you please explain what the values on both extremes of the graphs mean? They mostly seem to be happening around lag -24 and lag 24.

Technical suggestions:

1- I suggest signposting the paper early in the introduction. The reason is that the introduction although being an interesting read, is rather long.

2- It would be easier for the reader to have the zone numbers on the map in figure 1.

3- On page 7 line 19, it would be interesting to state briefly why the 2nd largest newspaper in the Aragon region selected, and not the 1st largest? Is it because unlike all the other newspapers in the region, it had an online record?

[Figure]

4- On page 8 line 9, why only "winter" cereal crops are selected?

5- On page 11, line 8, I think it would be an easier read if the lag example values appear with negative sign as they belong to the times before the central line.

6- I would have changed "this side of the central line" on page 11 line 8 to the "left" side of the central line to make it clearer.

7- Page 11, line 18, I think 0.6 should have a negative sign as it is referring to a negative correlation (as depicted in pink in figures 3 and 4).

8- On page 13 in the header, 3.3 is repeated twice.

―――――――――――――――――――――

---

## Author Comment (AC1) · 14 Mar 2017

We thank the reviewer for taking the time to review the manuscript thoroughly and for the helpful comments and suggestions. Here we provide answers to the specific comments and indications of how the manuscript has been improved to address the reviewer's concerns.

**Specific comment:**

**1- You have used the newspaper data as benchmark data for drought occurrence. Could you please elaborate on why you decided to choose newspaper**

**data? How reliable do you think the newspaper data is? What is the false alarm rate?**

The focus of our analysis was to identify indicators that could help managers in detecting drought conditions that may lead to impact and therefore we were looking for a benchmark based on drought impact occurrence.

The only available database of drought impact, the European Drought Impact report Inventory, does not provide enough records to allow an analysis at regional scale. And although there are a few reports and scientific articles that describe drought impacts in the area, especially in relation with the drought period 2005-2008, impact data in these studies is aggregated by year or even for the whole drought episode.

We selected newspaper records as a benchmark data source because it allowed us to systematically collect impact occurrence data of all affected sectors with a monthly time step for the whole period of analysis.

As mentioned in section 4.1. (second paragraph), a few reliability issues were noticed when collecting the data. The issues related to the wrong use of the word drought to report different situations such as summer shortages are the ones that may have the most impact on the reliability of the source. With this issue in mind, we classified the records of drought occurrence according to the source of the information about drought occurrence to make a distinction between official sources such as mandated authorities, managers and scientists (labelled in Fig 2 as "drought acknowledged by the authorities", "ongoing mitigation measures", and "periods retrospectively defined as anomalously dry", respectively) and non-official sources such as the journalist or the water users (labelled as "mention of drought occurrence"). This second type of source is the one that is susceptible to accuracy issues. In our case, only the mention of drought recorded in 2003 is not backed-up by mentions from official sources during the same period and may therefore be considered a false alarm. The rate of false records

derived from a misuse of the word drought depends on the rigour of the newspaper.

Bias issues (over or understatement of drought caused by political or public interests) do not have an influence in our analysis since we are only considering binary data of occurrence or non-occurrence. However, we acknowledge that these can have a significant impact on reliability if the records are used to estimate the severity of the event. These clarifications will be added to the second paragraph of the discussion.

**2- For the daily remote sensing data, you have used "monthly aggregate". Could you please explain more on how the aggregate was obtained?**

Monthly aggregates were generated from precipitation, Land Surface Temperature (LST) and Soil Moisture (SM) daily datasets. Precipitation data in mm/day was aggregated by a sum of the daily values for each pixel to obtain monthly data in mm/month. LST and SM data were aggregated by averaging the daily values for each pixel. A clarification will be added to the remote sensing data section.

**3- You have used the cross-correlation function as expressed by (Chatfield, 2004), which assumes the bivariate data is stationary and ergodic (on page 122 of Chatfiled, 2004). How did you test if your data was stationary and ergodic?**

We acknowledge that stationarity as well as ergodicity of the data is a point of concern. Part of the input data (LST, SM, NDVI, ET, GPP and PsNet) present a seasonal trend. For these, monthly anomalies were obtained by subtracting the mean for the whole period from each monthly average value, using these anomaly time series as input for the cross-correlation function. We have added a comment to clarify this in the manuscript.

To detect possible issues related to the stationarity or ergodicity of the series, their time autocorrelation and partial autocorrelation plots were considered. Overall, time plots show no trends or discontinuities and the values in the autocorrelation plots show that the autocorrelation diminishes quickly with increasing lag. An exception are the series of the reservoir indices. In that case, for some of the series it is not clear from the plot if the series is stationary. For one of them (management unit 122) it clearly is not. This management unit corresponds to a reservoir (Rialb) that started to be filled in the year 2000 and therefore the levels cannot be considered stationary for the period of study. Most of the autocorrelation plots for the reservoir level series present a small peak of autocorrelation at a lag of 12 months, and one of them (management unit 132) presents autocorrelation values declining more slowly (significant values until lag 20).

Additionally, we have run a stationarity test (Dickey-Fuller test) on the series. The results show that most of the datasets are stationary at the 95% confidence level. There are a few exceptions: for the LST series, two of the management units (120i and 130X) have a p-value of 0.07; while for the SPI-12 series, four of the management units have p-values higher than 0.05, these are management units 132 (p-value=0.07), 140 (0.10), 150 (0.19) and 151 (0.07); for the reservoir index series, again management units 122 and 132 present p-values over the significance level, 0.13 and 0.7 respectively.

We considered the remote sensing datasets to be stationary and ergodic enough to be used as input for the cross-correlation function. As for the reservoir data, a comment on the two reservoirs that do not satisfy the conditions could be included in the article.

We have verified the plots showing the anticipation to drought events. A small error was detected in these plots on careful review, which was corrected (Figure Q3 shows the corrected plots). In these plots we can clearly see that the two management units that do not satisfy the conditions for stationarity are those (at least two out of the three) that do not present anticipation. This last sentence will be included in the results section.
**4- You have used the cross-correlation function as a method of drought antici- pation. Can the same technique be applied for foreseeing the end of droughts?**

This is a very interesting question, given the difficulty in identifying the cessation of drought events. By using records of direct drought impact as a benchmark to assess the different parameters we were aiming to identify the onset of the conditions that cause these impacts. A good knowledge of the conditions that lead to impacts would be useful to identify both the start and end of a drought event, since the end of those conditions (return to normal conditions) would be an indicator of the end of the drought event.

**5- As you explained in figure 3 and 4, the values on the left side of (negative side) of the central line, show how early the remote sensing data have anticipated the newspaper headlines on droughts, or anticipated the decline in crop yield. Could you please explain what the values on the right side (positive side) of the central line show?**

The positive side of the plots reflects the correlation of the drought occurrence and impact series with the values of the different datasets in later months. For example at lag +1 the correlation between the impact and the value of the dataset the following month is shown. This type of correlation appears if the conditions that define the start of the event or impact occurrence last longer than one month.

For long term SPIs these correlation with positive lags are stronger since previous months are included in the SPI. For example for SPI-9 at lag +1 the drought event or impact occurrence is compared to an SPI built from the following month and the 8 previous months.

**6- Following from the previous comment, regarding figures 3 and 4, could you please explain what the values on both extremes of the graphs mean? They mostly seem to be happening around lag -24 and lag 24.**

The positive correlations of the timelines of drought occurrence and impacts with the values of the indicator datasets with lags over one year (most notably at -15 and $\pm24$ lags) are casual correlations. For example the figure Q6 shows how the periods of reported drought impacts coincide with positive anomalies when the data is shifted by a lag of -15.

Since we are focusing our evaluation at anticipations within a period of one hydrological year, the correlations should not be affected by this issue. An explanation will be added to the results section.

**Technical suggestions:**

**1- I suggest signposting the paper early in the introduction. The reason is that the introduction although being an interesting read, is rather long.**

We agree with the suggestion and we will introduce the aim of the paper at the end of the second paragraph (page 2, line 10) of the introduction instead of in the last paragraph (page 3, lines 25-27).

**2- It would be easier for the reader to have the zone numbers on the map in figure 1.**

We agree, and on inspection found that the figure seems to have lost the numbers

in the conversion to pdf. We have amended the figure to ensure these numbers are preserved.

**3- On page 7 line 19, it would be interesting to state briefly why the 2nd largest news-paper in the Aragon region selected, and not the 1st largest? Is it because unlike all the other newspapers in the region, it had an online record?**

Yes, the online record of the main newspaper only goes back to 2008. We also reviewed the available records for that newspaper, but did not include them in the analysis to avoid having the later years better represented than others. This will be clarified in the manuscript.

**4- On page 8 line 9, why only "winter" cereal crops are selected?**

Winter cereals are the cereal crops that are planted in the autumn and they are the crops that cover the largest surface area (especially barley and wheat). Their importance for the region results in better data availability than other crops and for this reason only this type of crops were selected for the analysis. A clarification will be added to the manuscript.

**Technical suggestions 5-8:**

We agree with the corrections suggested by the reviewer and will modify the text accordingly.

---

## Author Comment (AC2) · 15 Mar 2017

Annex to reply to reviewer 1 comments: Figure 1 (Question 3); Figure 2 (Question 6); Figure 3 (Suggestion 2)

[Figure]

[Figure]

**Fig. 1.** (Q3) Corrected status index cross-correlation plots

[Figure]

**Fig. 2.** (Q6) The dotted black line represents the anomaly of NDVI at zone 150r and the blue line is the same data with a lag of -15 months. The yellow and white points in the middle represent the months

[Figure]

**Fig. 3.** (S2) Study area map with numbered management units (corresponds to Figure 1 in the manuscript)

---

## Referee Comment (RC2) · Anonymous Referee #2 · 3 Apr 2017

This study analyses drought indicators' relation and thus usefulness to predict drought impacts for a small case study region in Spain. I find the study a worthy addition to the drought literature, once its contribution has been worked better. In the current manuscript the reader 'gets lost' in the many correlations and individual results a bit and I think, some focus and highlight is needed to better appreciate the work and results. The manuscript also requires clarifications on a number of methodological details and better justification of some parts of the approach in order to assess and value the results.

Major comments:

1) A particular concern I have is related to the use of a very short time period for the

'benchmarking' by correlation analysis. In this study, a time series from 2001 to 2012 is used with 8 out of these 12 years representing drought events (Section 3.1., beginning) – hence 2/3 of the time. The common understanding is, however, that drought is defined as a rare extreme climatic/hydrologic event. This has a few implications for the chosen method and conclusions from the study:

-To the international reader who doesn't know the Spanish climate history, it is not convincing whether simply 'dryness' or real 'drought' was analysed if 2/3 of the time were 'drought events' – how was this distinguished in the textual search? Does the Spanish language distinguish between the two? Some languages do, others don't.

- To define drought based on monitored hydro-meteorological or remotely sensed anomalies, long time-series are necessary to obtain the full range of situations and hence define the average and moments of distribution of the variable to be used to index drought as an extreme. How does that influence the results? The time series and temporal resolution for each index need to be given (inconsistent in current methods section), and where applicable the reference time series for standardization/normalization.

- Correlation depends on variability, but if two thirds of the time period are mostly dry, I would expect that this will have a considerable effect on the results of a statistic that relies on variability in the data. Some analysis regarding the sensitivity of the index series (and range of variability) on the results is therefore necessary to make an assessment of the uncertainties.

2) The objectives and used methods need to be better harmonized, in particular the relation of the use of statistical correlation analysis and the aimed for 'predictability' of impacts needs to be developed more clearly. An assessment of predictability of impacts would require a predictive application (some validation experiment). For hazards, it is also common to consider false alarm rates for an assessment of predictablity. Overall, the use of statistics and their interpretation in this study requires more clarification and

precision (see specific comments below).

3) The discussion section is rather vague in its comparison with other studies. In particular, there are many other studies that have correlated agriculture yields with drought indices in many countries (e.g. see table of the review by Bachmair et al., 2016 in WIRES Water). How do the findings for best-correlated indices and time scales compare to other studies? If this case study wants to contribute to the international literature these need to be better compared in the discussion. The niche of this study within the wider range of studies needs to be worked out more specifically to appreciate this small scale case study's contribution to the field.

Specific comments

Methods section:

4) The twofold use and steps of impact report analyses, (1)construct a narrative of the events, (2) use as binary indicator of impact in correlation analysis, should also be introduced as such in the methods section. The narrative in the first part of the results otherwise comes really unexpected, whereas the reader expects only correlation results.

5) P1 Last paragraph: another reason that RS data needs benchmarking are the short time series – compared to precipitation and hydrometric records a definition of drought as statistical extreme is not possible. THis requires elaboration on assumptions made and limitations for the analysis.

6) P8 lines 3-5. It is necessary to add some justification for the categorizations of sectors and of type of information. Sectors: e.g. readers may wonder why only rainfed 'cereal' – are there no other rainfed crops? Is all irrigated agriculture similar regarding seasonality/demand? Type: why were these distinguished and e.g. what would be different whether a drought is retrospectively reported or "mentioned". I couldn't find how this classification was used in the correlation analyses – if it wasn't used in the

analyses, it is not relevant to mention in the methods section.

7) p.8 line 14 ff. From that section it is unclear, which variables were correlated with which exactly. I suggest to name all variables in the data section and then here add a clear list/matrix of what to expect in the results section.

8) P8 line 15 Isn't it the other way round: predictability may be (! But not necessarily) related to the strength of the relation ..... Please elaborate more precisely the link between a correlation analysis and it's potential for prediction (of what exactly?) – this is not necessarily methodologically straightforward.

9) p.8 line 28. How is binary information used in the calculation of correlations, which normally require ordinally scaled data. How is significance tested? How is auto-correlation corrected/considered (Spain always emphasizes that they have multi-year droughts). Citing an R function cannot replace a complete introduction of the statistical methods used, incl. all variable transformations and an introduction of all the measures used later in Results and Discussion sections.

10) P9 line 13-end belongs into the methods section. Now it is partly a 1:1 repetition (unnecessary) and more explanation on the method than in the methods section – should be the other way round.

Results section

11) Similar to the impact reports, some time series of the longer series standardized indices (and RS variables?) need to be shown to support that we are indeed looking at drought as an extreme event (e.g. also in 3.1).

12) p.11 line6ff – a description of figure legend in the text is not necessary. This info should be clear from methods section/variable definition and figures' legends and captions. Instead, a more detailed description of correlation patterns shown by the figures needs to be given to warrant having them all in the paper.

12) Quite a bit of discussion is already included in the results section, which makes

it a bit difficult for the reader to see the main outcomes, i.e. the strengths of correlations found, the differences and similarities, etc... without immediately being biased towards a possible background explanation. I suggest to moved all sentences with references into the discussion section and here purely describe own results first (see also introductory comment on better focus on unique results).

Figures

13) Fig. 3 (and similar) The heading is misleading. Many readers would expect scatterplots of indicators vs impacts. The legend needs a header relating to a clear variable from the methods section (maximum of cross-correlation what - coefficient? What exactly is shown?)

14) Figure 5 (and similar) is very hard to see/read as labels are too small on A4 print. Also captions and legend labels need to be more precise (p-Values for which test – methods section?) shown are circles scaled to p-values (range not one value), but not "significances" (better to define previously).

Discussion section

13) The discussion misses a critical evaluation of the statistics used with respect to assumptions on methods and data (besides the impact reports). In particular, more insight into the RS data and what they inherently 'see' or not linked to drought impacts would help advance the selection of future usage of these as indices.

14) p. 18 line 17ff. The list of caveats is good, but will be more useful if it were expanded by how these will actually each affect the results.

Suggestion: For consideration of seasonal aspects and time trends in impact occurrence modelling, perhaps the discussion by Stagge et al. 2015 may provide some further ideas.

References

[Figure]

Bachmair, S., Stahl, K., Collins, K., Hannaford, J., Acreman, M., Svoboda, M., Knutson, C., Smith, K., Wall, N., Fuchs, B., Crossman, N., Overton, I. (2016) Drought indicators revisited: the need for a wider consideration of environment and society Wiley Interdisciplinary Reviews: Water, 3: 516–536. DOI: 10.1002/wat2.1154

Stagge, J.H., Kohn, I., Tallaksen, L.M., Stahl, K. (2015) Modeling drought impact occurrence based on meteorological drought indices in Europe Journal of Hydrology 530, 37–50 doi.:10.1016/j.jhydrol.2015.09.039

---

## Author Comment (AC3) · 4 May 2017

We thank the reviewer for providing detailed comments that helped us to further reflect on the analysis presented in the manuscript. Our replies to those comments are included below, together with indications on how we propose to improve the manuscript based on the suggestions of the reviewer.

Note: The numbering of the comments has been modified to fix some existing repetitions in the original numbering.

......................................................................................................

This study analyses drought indicators' relation and thus usefulness to predict drought impacts for a small case study region in Spain. I find the study a worthy addition to the drought literature, once its contribution has been worked better. In the current manuscript the reader 'gets lost' in the many correlations and individual results a bit and I think, some focus and highlight is needed to better appreciate the work and results. The manuscript also requires clarifications on a number of methodological details and better justification of some parts of the approach in order to assess and value the results.

**Major comments:**

1) A particular concern I have is related to the use of a very short time period for the 'benchmarking' by correlation analysis. In this study, a time series from 2001 to 2012 is used with 8 out of these 12 years representing drought events (Section 3.1., beginning) - hence 2/3 of the time. The common understanding is, however, that drought is defined as a rare extreme climatic/hydrologic event. This has a few implications for the chosen method and conclusions from the study:

Our aim in the analysis presented in the manuscript was to assess the usefulness of medium resolution global remote sensing products for drought management at basin scale. The length of the time series used in the analysis is therefore limited by the availability of those remote sensing products. We acknowledge that 12 years is a short period and that this limits the variability represented in the series. In fact our series includes three drought events, one of them having impacts over four hydrological

years. The short length of the data series available was one of the reasons to base the definition of drought we use to build the reference not on a frequency analysis, in which drought is defined as an extreme event with respect to the historical series, but on the occurrence of drought impacts. The other reason being that what managers need is to identify the conditions that may lead to drought impacts in order to take mitigation actions. For these two reasons we defined drought in the analysis as conditions of meteorological origin that may lead to impacts in sectors depending on water and we used impact data as benchmark information. This will be clarified in the manuscript in section 2.2.3 (Benchmarking datasets).

**a.- To the international reader who doesn't know the Spanish climate history, it is not convincing whether simply 'dryness' or real 'drought' was analysed if 2/3 of the time were 'drought events' – how was this distinguished in the textual search? Does the Spanish language distinguish between the two? Some languages do, others don't.**

The Spanish language does have different words for dryness (sequedad) and drought (sequía). The term 'sequía' was used for the textual search. As we commented in the discussion, the possible misuse of the word drought/sequía in the press was one of our concerns when reviewing the news. This was the reason to differentiate the records that are just mentions of drought, from others that are reporting the acknowledgement of drought from an official source. The classified records in figure 2 of the manuscript (upper part) show that only one of the records (the mention of drought recorded in 2003) is not backed-up by the acknowledgement of drought from official sources during the same period, and therefore may be considered a case of misuse of the word. We will include a specific comment in the discussion about the accuracy of the impact data (p.18, line 17).

**HESSD**

The drought events identified by the textual search for a sector of the Ebro basin correspond with events observed at a larger scale. For example, Spinoni et al. (2015), use an indicator that combines three precipitation and potential evaporation-based indices to identify the drought events occurred in different regions of Europe during the period 1950-2012. Following that approach they identify three drought events for the Iberian Peninsula for the period 2000-2012 that match the ones obtained by the textual search, with the difference that the event starting in the hydrological year 2004-2005 has a shorter duration. This is caused by the different spatial scale of the analysis. While most of the basins in Spain received normal precipitation during the hydrological year 2006-2007, in the Ebro basin, and especially in the inner part of Cataluña, it was still low during that year (MMA, 2007). The reference to the work of Spinoni, corroborating our results in identifying three drought events during this short period will be also included in the discussion about the reliability of the impact data (after the second paragraph in section 4.1).

**b.- To define drought based on monitored hydro-meteorological or remotely sensed anomalies, long time-series are necessary to obtain the full range of situations and hence define the average and moments of distribution of the variable to be used to index drought as an extreme. How does that influence the results? The time series and temporal resolution for each index need to be given (inconsistent in current methods section), and where applicable the reference time series for standardization/normalization.**

Our drought reference is based on the occurrence of impacts, rather than on the identification of extremes in the time series. This is explained in more detail above, in the first paragraph of the reply to this comment, and will be also clarified in the

manuscript in section 2.2.3 (Benchmarking datasets).

The length of the period of analysis doesn't have an influence in the identification of drought events based on impact records. However, having a longer series, and therefore potentially a larger number of drought events, would provide more robust results in the correlation analysis. Ideally the results should be updated as the period of record of remote sensing data grows. The implications of the length of the remote sensing series will be added to the discussion in section 4.2.

We have verified the information provided in the manuscript regarding the temporal resolution of the datasets and have realised that the reference to the monthly GPP and PsnNet product used in the analysis, downloaded from the Numerical Terradynamic Group (NTSG), is missing. This will be added to the text (p.6, line 16). The temporal resolution of all the selected products is shown in Table 1 in the column 'Original time interval'. No additional data has been used as a reference for standardization/normalization.

**c.- Correlation depends on variability, but if two thirds of the time period are mostly dry, I would expect that this will have a considerable effect on the results of a statistic that relies on variability in the data. Some analysis regarding the sensitivity of the index series (and range of variability) on the results is therefore necessary to make an assessment of the uncertainties.**

The correlation analysis is performed with a monthly time step. According to the news review, during the period of analysis we have drought conditions occurring 63 months out of 136 (46%) and impacts 58 months out of 136 (42%). These records

correspond to three drought events. One of them is a multiyear event of extreme severity (2004-2008). In fact, the hydrological year 2004-2005 was characterised as one of the most intense droughts of the record in the Iberian Peninsula (García-Herrera et al., 2007). The other two events affect a single hydrological year at a smaller scale. However, despite the shortness of the period (12 years), it encompasses a wide range of different conditions, including also one of the wettest hydrological years of the country's record, the hydrological year 2003-2004 (MMA, 2005). The variability of the period will be mentioned in the description of the study area.

We acknowledge that a sensitivity test would contribute to reinforce the validity of the results. However, a longer series than the one available would be necessary to ensure an adequate estimation of the sensitivity. A detailed discussion on how the significance of the results and the possible issues of the remote sensing series were considered in the analysis is provided in the reply to comment 9.

**2) The objectives and used methods need to be better harmonized, in particular the relation of the use of statistical correlation analysis and the aimed for 'predictability' of impacts needs to be developed more clearly. An assessment of predictability of impacts would require a predictive application (some validation experiment). For hazards, it is also common to consider false alarm rates for an assessment of predictablity. Overall, the use of statistics and their interpretation in this study requires more clarification and precision (see specific comments below).**

The analysis presented in the manuscript can be considered a preliminary study on drought prediction. Indeed, we agree that the development of an operational detection method or tool would require further analysis and validation. At this initial stage,

[Figure]

however, our aim is to identify the datasets that can be useful for operational drought detection at the basin scale. Drought detection in this case is closely related to the predictability of impacts, as the conditions that need to be detected are those that may lead to impacts. However, these impacts do not necessarily occur immediately; their occurrence can be delayed as the effects of drought propagate through the different components of the hydrological cycle. To identify the remote sensing parameters that represent conditions that anticipate the occurrence of drought impacts, and therefore have potential to support the prediction of drought, we explore the correlation between the remote sensing data and the drought impacts at different time lags. While using correlation in this way may say less about the long term correlation of two time series, it does provide insight in the relationship between correlation and lag. We will include a clarification of the aim of the analysis in section 2.3 (description of the correlation analysis).

**3) The discussion section is rather vague in its comparison with other studies. In particular, there are many other studies that have correlated agriculture yields with drought indices in many countries (e.g. see table of the review by Bachmair et al., 2016 in WIRES Water). How do the findings for best-correlated indices and time scales compare to other studies? If this case study wants to contribute to the international literature these need to be better compared in the discussion. The niche of this study within the wider range of studies needs to be worked out more specifically to appreciate this small scale case study's contribution to the field.**

We agree that the correlation of remote sensing data, especially SPI and NDVI, to agriculture yield data has been widely researched and applied. Rather than aiming to add something new to that field, the purpose of including that part was to provide

a comparison of the results obtained in the correlation to text based impact data and results obtained with the most commonly used type of impact data, crop yields, and discuss the advantages and limitations of one with respect to the other. An explanation of this purpose and references to the wide use of correlation between crop yields and drought indices will be added to section 2.3 (description of the correlation analysis). Additional references and comparison with other studies will be also added as mentioned in the replies to comment 1a and to the reference suggestion at the end of this review.

**Specific comments**

**Methods section:**

**4) The twofold use and steps of impact report analyses, (1)construct a narrative of the events, (2) use as binary indicator of impact in correlation analysis, should also be introduced as such in the methods section. The narrative in the first part of the results otherwise comes really unexpected, whereas the reader expects only correlation results.**

We agree and will mention the use of the impact records to reconstruct the onset and evolution of drought conditions during the period of analysis in section 2.2.3 (Benchmark datasets).

**5) P1 Last paragraph: another reason that RS data needs benchmarking are the short time series - compared to precipitation and hydrometric records**

**a definition of drought as statistical extreme is not possible. This requires elaboration on assumptions made and limitations for the analysis.**

As mentioned in the reply to comment 1b, the analysis was not based on a definition of drought as a statistical extreme, but as the occurrence of certain conditions of meteorological origin that lead to impacts in sectors depending on water. This will be clarified in the last paragraph of the introduction and the implications for the results described in the reply to comment 1b (second paragraph) will be included in the discussion.

**6) P8 lines 3-5. It is necessary to add some justification for the categorizations of sectors and of type of information. Sectors: e.g. readers may wonder why only rainfed 'cereal' - are there no other rainfed crops? Is all irrigated agriculture similar regarding seasonality/demand? Type: why were these distinguished and e.g. what would be different whether a drought is retrospectively reported or "mentioned". I couldn't find how this classification was used in the correlation analyses - if it wasn't used in the analyses, it is not relevant to mention in the methods section.**

We agree that 'rainfed agriculture' is a better label for that sector than 'rainfed cereal' and we will substitute it in the text and figure 2.

The main types of irrigated crops in the area are fruit orchards, alfalfa and maize and they indeed have different seasonality and water demand. The irrigation campaign typically runs from March to October. Most of the records classified as impacts to irrigated agriculture in the analysis refer to insufficient water available for irrigation

and curtailments to the irrigation quota. This is a situation that will affect all crops that require water during the period of the curtailment, although certain crops may be more affected depending on the growing stage at the time of the curtailment.

As mentioned in the reply to comment 1a, the reason to make a distinction between the records that are just mention of drought from others that are reporting the acknowledgement of drought from an official source is the different level of reliability attributed to the two types of records. This distinction was not used in the correlation analysis, but was considered in the interpretation of the results.

**7) p.8 line 14 ff. From that section it is unclear, which variables were correlated with which exactly. I suggest to name all variables in the data section and then here add a clear list/matrix of what to expect in the results section.**

The correlation is performed between each of the remote sensing parameters and both the timeline that aggregates all types of drought events records and the timeline that aggregates all types of drought impacts (figure 2 in the manuscript). We will rephrase the first sentence of the section 2.3 to make it clearer.

**8) P8 line 15 Isn't it the other way round: predictability may be (! But not necessarily) related to the strength of the relation ..... Please elaborate more precisely the link between a correlation analysis and it's potential for prediction (of what exactly?) - this is not necessarily methodologically straightforward.**

With the sentence 'The strength of the relationship is related to the predictability of

the occurrence of drought and drought impacts provided by the remote sensing time series' we wanted to point out that the two things are connected. If the remote sensing product does provide information on the occurrence of drought or drought impacts the correlation observed should be stronger. This occurrence of drought impacts is what we are most interested in predicting, since the managers require this information to apply measures to mitigate those possible impacts. When analysing the results of the correlation we need to take into account the limitations of the test before we can establish causal relations between the correlation values obtained and the predictive capability of the product. We will substitute 'is related to' with 'is a function of' in that sentence to avoid the ambiguity.

**9) p.8 line 28. How is binary information used in the calculation of correlations, which normally require ordinally scaled data. How is significance tested? How is auto-correlation corrected/considered (Spain always emphasizes that they have multi-year droughts). Citing an R function cannot replace a complete introduction of the statistical methods used, incl. all variable transformations and an introduction of all the measures used later in Results and Discussion sections.**

For the correlation analysis between the remote sensing parameters (continuous variable) and the data of drought (impact) occurrence or non-occurrence (dichotomous categorical variable) Pearson correlations were calculated by assigning a numeric binary code to the categories of the dichotomous variable. In this case 1 was assigned to drought (impact) occurrence and 0 to the non-occurrence. The correlation between a continuous and a dichotomous variable is sometimes called point-biserial correlation, but the formula is mathematically equivalent to that of Pearson correlation (Cohen, Cohen, West, & Aiken, 2003).

Testing the significance of cross-correlation results is indeed a delicate issue. The bounds provided by the function in R are just rough guidelines and must be interpreted carefully. The bounds provided in this case are below 0.2 and above -0.2 and for this reason values between -0.2 and 0.2 were not considered for the plots. In view of the difficulty to calculate exact boundaries to test if the cross-correlation results are significantly different from 0, Chatfield (2004) suggests leaning on graphic or numeric tools to support the results.

The time, autocorrelation and partial autocorrelation plots for the series were explored to detect possible issues with the data. The autocorrelation plots show that the auto-correlation diminishes quickly with increasing lag, with the exception of the reservoir indices series. Most of the autocorrelation plots for the reservoir level series present a small peak of autocorrelation at a lag of 12 months, and one of them (management unit 132) presents autocorrelation values declining more slowly (significant values until lag 20).

For the remaining products, autocorrelation for ET, LST, GPP, PsnNet, SM and SPI-3 dissipates mostly at a lag of 2 months. For SPI-1 it goes quicker and is non-existent in some cases. NDVI takes 3-4 months and for SPIs with longer accumulation periods (SPI-6, 9 and 12) the correlation dissipates slower (4, 6 and 8 months respectively), which is inherent to the product. This can indeed have an influence on the correlation of the impacts timeline to NDVI and SPI-6 (and longer), showing stronger correlations. This will be emphasised in the discussion section.

The R function used performs just the cross-correlation function, which we consider to be a well-known function documented in many reference statistic texts. One of these

references was cited in the manuscript. We will rephrase the sentence to have the reference text citation before the mention to the R function. We will also add to this section the explanation regarding the purpose of the correlation test between crop yields and drought indices as described in the reply to comment 3.

**10) P9 line 13-end belongs into the methods section. Now it is partly a 1:1 repetition (unnecessary) and more explanation on the method than in the methods section - should be the other way round.**

We agree and will remove that paragraph from the results section and complete the explanation in the methods section.

**Results section**

**11) Similar to the impact reports, some time series of the longer series standardized indices (and RS variables?) need to be shown to support that we are indeed looking at drought as an extreme event (e.g. also in 3.1).**

We are basing the definition of drought for the analysis on impact occurrence rather than on a frequency analysis, in which drought is defined as an extreme event with respect to the historical series. This is explained in more detail in the reply to comment 1. To put the variability of the test period in the context of the historic record we will include additional references in the section that describes the basin area as mentioned in the reply to comment 3.

**12) p.11 line6ff - a description of figure legend in the text is not necessary. This info should be clear from methods section/variable definition and figures' legends and captions. Instead, a more detailed description of correlation patterns shown by the figures needs to be given to warrant having them all in the paper.**

Other reviewers, prior to the submission of the manuscript, considered it useful to have a written description of those plots since they found them uncommon. But we will remove the sentence about the x-axis (line 5) as it is already mentioned in the caption of the figure.

We will also add more detail about the correlation patterns as suggested. Especially regarding the differences between rainfed and irrigated areas that appear for some of the products.

**13) Quite a bit of discussion is already included in the results section, which makes it a bit difficult for the reader to see the main outcomes, i.e. the strengths of correlations found, the differences and similarities, etc... without immediately being biased towards a possible background explanation. I suggest to moved all sentences with references into the discussion section and here purely describe own results first (see also introductory comment on better focus on unique results).**

We agree and will move the explanations regarding the possible reason for weaker correlations for soil moisture (p.11, line 20-22) and GPP-PsnNet (p.11, line 33 - p12)

and of stronger correlations for ET and NDVI (p. 15, line 2 - p. 16, line 1) to the discussion section.

The mention to local press reporting drought impacts of the hydrological year 2006-2007 that were not reflected in the regional press (p. 17, lines 5-7), will be also moved to the discussion section as an example of scale issues (p.18, line 24).

**Figures**

**14) Fig. 3 (and similar) The heading is misleading. Many readers would expect scatter-plots of indicators vs impacts. The legend needs a header relating to a clear variable from the methods section (maximum of cross-correlation what - coefficient? What exactly is shown?)**

The headers of figures 3 and 4 will be removed, since the captions already define the plots as the cross-correlation of drought indicators and drought events or impacts.

The symbol for the cross-correlation (defined in equation 4) will be added in the legend. We have found that the formulas for the cross-covariance and correlation included in the manuscript correspond to the theoretical cross-covariance and correlation, rather than the sample cross-covariance and correlation. The former will be substituted by the latter, which is identical but with Latin instead of Greek letters. The header for the legend on figures 3 and 4 will then be $r_{xy}$.

**15) Figure 5 (and similar) is very hard to see/read as labels are too small on A4 print. Also captions and legend labels need to be more precise (p-Values for which test - methods section?) shown are circles scaled to p-values (range not one value), but not "significances" (better to define previously).**

The font size in figures 5 and 6 has been increased and the p-values in the legend have been substituted by intervals. The p-values in the figures correspond to the correlation test and refer to the probability to obtain those (or more extreme) results if the variables were not correlated. We will change the word 'significance' to 'reliability' in the caption to avoid the ambiguity.

**Discussion section**

**16) The discussion misses a critical evaluation of the statistics used with respect to assumptions on methods and data (besides the impact reports). In particular, more insight into the RS data and what they inherently 'see' or not linked to drought impacts would help advance the selection of future usage of these as indices.**

This is related with the issues discussed in the reply to comment 9 in relation to testing the significance of the results and, most notably, with the implications of the autocorrelation length of the different parameters. As discussed above the latter can have an influence in the strength of the correlation obtained for NDVI, especially for SPI-6 and longer. We will include this in the discussion section, together with the discussion of the results that was originally included in the results section (see reply to comment 13).

**17) p. 18 line 17ff. The list of caveats is good, but will be more useful if it were expanded by how these will actually each affect the results.**

We agree that this is a useful addition to the discussion. The effects on the results observed for the different aspects included in the list of caveats are the following:

– Accuracy (related to the misuse of the word drought in the newspaper): As previously mentioned in the reply to comment 1a, this issue was the reason to classify the records of drought occurrence according to the source of the information to make a distinction between official sources such as mandated authorities, managers and scientists; and non-official sources such as journalists or water users. This second type of source is the one that is most susceptible to accuracy issues. Particularly for the case of the mandated authorities there are clear procedures with which drought is officially acknowledged, which are defined in the drought management plan. In the records reviewed, only the mention of drought conditions recorded in 2003 is not backed-up by mention of drought from official sources during the same period, and may therefore be regarded as a misuse of the word. Thus, we consider accuracy issues to have little impact on results.

– Completeness: Certain types of impacts received a wider and more detailed coverage. In Fig. 2 of the manuscript some unlikely situations can be identified. For example impacts to livestock in May and July 2006, but not in June. The effect of the incompleteness is reduced by aggregating all reports of impacts across different sectors.

– Scale: The results of the test with crop yield data show values for the hydrological year 2006-2007 for which no drought impacts were identified in the reviewed

regional newspaper that are similar to three other hydrological years for which drought impacts were recorded. Local press for the specific area of the test (Alto Aragón), however, reported a lack of rain from October to March, aggravated with high temperatures, in Monegros and Bajo Cinca that had an impact on rainfed cereals and pastures.

– Bias (related to the over or understatement of drought caused by political or public interests): These do not have an influence in our analysis since we are only considering binary data of occurrence or non-occurrence, but this issue could have a significant impact on the reliability if the records were used to estimate the severity of the event.

These effects will be added to the list of caveats.

**Suggestion: For consideration of seasonal aspects and time trends in impact occurrence modelling, perhaps the discussion by Stagge et al. 2015 may provide some further ideas.**

Thank you for the relevant reference. It does provide additional insights in the topic and the methodological approach based on logistic regression and Generalized Additive Models (GAMs) is worth considering for further steps. We will reference it in the introduction section as an example of use of impact data to assess indicators for drought detection (p. 3, line 14). It is also a good reference to illustrate in the discussion how the most informative indicators of drought occurrence may vary depending on specific characteristics of the country or basin, such as management practices or dominant water uses.

**References**

**Bachmair, S., Stahl, K., Collins, K., Hannaford, J., Acreman, M., Svoboda, M., Knutson, C., Smith, K., Wall, N., Fuchs, B., Crossman, N., Overton, I. (2016) Drought indicators revisited: the need for a wider consideration of environment and society Wiley Interdisciplinary Reviews: Water, 3: 516-536. DOI: 10.1002/wat2.1154**

**Stagge, J.H., Kohn, I., Tallaksen, L.M., Stahl, K. (2015) Modeling drought impact occurrence based on meteorological drought indices in Europe Journal of Hydrology 530, 37-50 doi.:10.1016/j.jhydrol.2015.09.039**

Chatfield, C. (2004). Time-series forecasting. New York: Chapman Hall/CRC.

Cohen, J., Cohen, P., West, S. G., & Aiken, L. S. (2003). Applied multiple regression/correlation analysis for the behavioural sciences (3rd ed.). Mahwah, New Jersey: Lawrence Erlbaum Associates.

García-Herrera, R., Paredes, D., Trigo, R. M., Franco Trigo, I., Hernández, E., Barriopedro, D., & Mendes, M. A. (2007). The Outstanding 2004/05 Drought in the Iberian Peninsula: Associated Atmospheric Circulation. American Meteorological Society, 8, 483-498.

MMA. (2005). Informe balance del año hidrológico 2004-2005 (Assessment of the hydrological year 2004-2005) (Programa A.G.U.A.) (p. 141). Madrid (Spain): Ministerio de Medio Ambiente.

Spinoni, J., Naumann, G., & Vogt, J. V. (2015). The biggest drought events in Europe from 1950 to 2012. Journal of Hydrology: Regional Studies, 3, 509-524. http://doi.org/10.1016/j.ejrh.2015.01.001

---

## Author Response (AR2)

A few changes were made to the manuscript to merge short paragraphs, the name of the institute was updated in the affiliation and text, and a few orthographic/style corrections were made. A marked up-manuscript follows.

......................................................................................................................

[revised manuscript text omitted]

Bachmair, S., Svensson, C., Hannaford, J., Barker, L.J. & Stahl, K. (2016). A quantitative analysis to objectively appraise drought indicators and model drought impacts. *Hydrology and Earth System Sciences 20*(7): 2589–2609.

Beguería, S. & Vicente-Serrano, S.M. (2013). *SPEI: Calculation of the Standardised Precipitation-Evapotranspiration Index. R package version 1.6.* Retrieved from https://CRAN.R-project.org/package=SPEI

Blauhut, V., Gudmundsson, L. & Stahl, K. (2015). Towards pan-European drought risk maps: quantifying the link between drought indices and reported drought impacts. *Environmental Research Letters 10*: 10.

Brown, J.F., Wardlow, B.D., Tadesse, T., Hayes, M.J. & Reed, B.C. (2008). The Vegetation Drought Response Index (VegDRI): A new integrated approach for monitoring drought stress in vegetation. *GIScience & Remote Sensing 45*(1): 16–46.

Chatfield, C. (2004). *Time-series forecasting*. New York: Chapman & Hall/CRC.

CHE. (2007). *Plan Especial de Actuación en Situaciones de Alerta y Eventual Sequía en la Cuenca Hidrográfica del Ebro*. Confederación Hidrográfica del Ebro. Retrieved from http://www.chebro.es/contenido.streamFichero.do?idBinario=5889

CHE. (n.d.). Portal de CHEbro. Retrieved from http://www.chebro.es/

Chen, M., Senay, G.B., Singh, R.K. & Verdin, J.P. (2016). Uncertainty analysis of the Operational Simplified Surface Energy Balance (SSEBop) model at multiple flux tower sites. *Journal of Hydrology*.

CHJ. (2007). *Plan especial de alerta y eventual sequía en la Confederación Hidrográfica del Júcar*. Confederación Hidrográfica del Júcar. Retrieved from http://www.chj.es/es-es/medioambiente/gestionsequia/Paginas/PlanEspecialdeAlertayEventualSequia.aspx

Commission of the European Communities. Communication from the Commission to the European Parliament and the Council - Addressing the challenge of water scarcity and droughts in the European Union. , Pub. L. No. COM(1007) 414 final (2007). Retrieved from http://eur-lex.europa.eu/legal-content/EN/TXT/PDF/?uri=CELEX:52007DC0414&from=EN

Famiglietti, J.S., Cazenave, A., Eicker, A., Reager, J.T., Rodell, M., Velicogna, I., … Zhulidov, A.V. (2015). Watching water: From sky or stream? *Science 349*(6249): 684.

Funk, C., Peterson, P., Landsfeld, M., Pedreros, D., Verdin, J., Shukla, S., … Michaelsen, J. (2015). The climate hazards infrared precipitation with stations—a new environmental record for monitoring extremes. *Scientific Data 2*.

García-Herrera, R., Paredes, D., Trigo, R.M., Franco Trigo, I., Hernández, E., Barriopedro, D. & Mendes, M.A. (2007). The Outstanding 2004/05 Drought in the Iberian Peninsula: Associated Atmospheric Circulation. *American Meteorological Society 8*: 483–498.

Guerschman, J.P., Van Dijk, A.I.J.M., Mattersdorf, G., Beringer, J., Hutley, L.B., Leuning, R., … Sherman, B.S. (2009). Scaling of potential evapotranspiration with MODIS data reproduces flux observations and catchment water balance observations across Australia. *Journal of Hydrology 369*(1–2): 107–119.

Hain, C.R., Mecikalski, J.R. & Anderson, M.C. (2009). Retrieval of an Available Water-Based Soil Moisture Proxy from Thermal Infrared Remote Sensing. Part I: Methodology and Validation. *Journal of Hydrometeorology 10*(3): 665–683.

Hernández Varela, L., Lozano Valencia, M.Á. & Soleto García, C. (2003). Estudio de los acontecimientos meteorológicos extraordinarios en la comunidad autónoma del País Vasco (1870-1954) a través de la prensa. *Investigaciones Geográficas 30*: 165–180.

Hernández-Mora, N., Gil, M., Garrido, A. & Rodríguez-Casado, R. (2013). *La sequía 2005-2008 en la cuenca del Ebro. Vulnerabilidad, Impactos y Medidas de Gestión*. Universidad Politécnica de Madrid - CEIGRAM.

Iglesias, A., Garrote, L. & Martín-Carrasco, F. (2009). Drought risk management in mediterranean river basins. *Integrated Environmental Assessment and Management 5*(1): 11–16.

Kallis, G. (2008). Droughts. *Annual Review of Environment and Resources 33*: 85–118.

Keyantash, J. & Dracup, J.A. (2002). The Quantification of Drought: An Evaluation of Drought Indices. *Bulletin of the American Meteorological Society 83*(8): 1167–1180.

Kogan, F.N. (2001). Operational Space Technology for Global Vegetation Assessment. *Bulletin of the American Meteorological Society 82*(9): 1949–1964.

Lackstrom, K., Brennan, A., Ferguson, D., Crimmins, M., Darby, L., Dow, K., … Smith, K. (2013). *The Missing Piece: Drought Impacts Monitoring* (Report from a Workshop in Tucson, AZ). Carolinas Integrated Sciences & Assessments program and the Climate Assessment for the Southwest.

Liu, Y.Y., Dorigo, W.A., Parinussa, R.M., de Jeu, R.A.M., Wagner, W., McCabe, M.F., … van Dijk, A.I.J.M. (2012). Trend-preserving blending of passive and active microwave soil moisture retrievals. *Remote Sensing of Environment 123*: 280–297.

Liu, Y.Y., Parinussa, R.M., Dorigo, W.A., De Jeu, R.A.M., Wagner, W., van Dijk, A.I.J.M., … Evans, J.P. (2011). Developing an improved soil moisture dataset by blending passive and active microwave satellite-based retrievals. *Hydrol. Earth Syst. Sci. 15*(2): 425–436.

Llasat, M.C., Llasat-Botija, M., Barnolas, M., López, L. & Altava-Ortiz, V. (2009). An analysis of the evolution of hydrometeorological extremes in newspapers: the case of Catalonia, 1982-2006. *Natural Hazards and Earth System Sciences 9*: 1201–1212.

McKee, T.B., Doesken, N.J. & Kleist, J. (1993). The relationship of drought frequency and duration to time scales. In *Proceedings of the 8th Conference on Applied Climatology* (Vol. 17). American Meteorological Society Boston, MA.

Miralles, D.G., Holmes, T.R.H., de Jeu, R.A.M., Gash, J.H., Meesters, A.G.C.A. & Dolman, A.J. (2011). Global land-surface evaporation estimated from satellite-based observations. *Hydrology and Earth System Sciences 15*: 453–469.

MMA. (2005). *Informe balance del año hidrológico 2004-2005 (Assessment of the hydrological year 2004-2005)* (Programa A.G.U.A.). Madrid (Spain): Ministerio de Medio Ambiente. Retrieved from http://www.mapama.gob.es/ministerio/pags/Biblioteca/Revistas/pdf_IBH%2FBAH _2004_05.pdf

Morid, S., Smakhtin, V. & Moghaddasi, M. (2006). Comparison of seven meteorological indices for drought monitoring in Iran. *International Journal of Climatology 26*(7): 971–985.

Mu, Q., Heinsch, F.A., Zhao, M. & Running, S.W. (2007). Development of a global evapotranspiration algorithm based on MODIS and global meteorology data. *Remote Sensing of Environment 111*(4): 519–536.

Mu, Q., Zhao, M. & Running, S.W. (2011). Improvements to a MODIS global terrestrial evapotranspiration algorithm. *Remote Sensing of Environment 115*: 1781–1800.

Perez y Perez, L. & Barreiro-Hurlé, J. (2009). Assessing the socio-economic impacts of drought in the Ebro River Basin. *Spanish Journal of Agricultural Research 7*(2): 269–280.

Potop, V. (2011). Evolution of drought severity and its impact on corn in the Republic of Moldova. *Theoretical and Applied Climatology 105*(3): 469–483.

Redmond, K.T. (2002). The depiction of drought - A commentary. *Bulletin of the American Meteorological Society 83*: 1143–1147.

Running, S.W. & Zhao, M. (2015, October 7). User's Guide - Daily GPP and Annual NPP (MOD17A2/A3) Products NASA Earth Observing System MODIS Land Algorithm (version 3.0). Retrieved from http://www.ntsg.umt.edu/sites/ntsg.umt.edu/files/modis/MOD17Users Guide2015_v3.pdf

Scott, C.A., Bastiaanssen, W.G.M. & Ahmad, M.-D. (2003). Mapping Root Zone Soil Moisture Using Remotely Sensed Optical Imagery. *Journal of Irrigation and Drainage Engineering 129*(5): 326–335.

Senay, G.B., Bohms, S., Singh, R.K., Gowda, P.H., Velpuri, N.M., Alemu, H. & Verdin, J.P. (2013). Operational Evapotranspiration Mapping Using Remote Sensing and Weather Datasets: A New Parameterization for the SSEB Approach. *Journal of American Water Resources Association 49*(3): 577–591.

Sepulcre, G., Horion, S.M.A.F., Singleton, A., Carrao, H. & Vogt, J. (2012). Development of a Combined Drought Indicator to detect agricultural drought in Europe. *Natural Hazards and Earth System Sciences 12*(11): 3519–3531.

Sheffield, J., Wood, E.F., Chaney, N., Guan, K., Sadri, S., Yuan, X., … Ogallo, L. (2014). A drought monitoring and forecasting system for sub-Sahara African water resources and food security. *American Meteorological Society* 861–882.

Smakhtin, V.U. & Schipper, L.F. (2008). Droughts: The impact of semantics and perceptions. *Water Policy 10*: 131–143.

Sonnett, J., Morehouse, B.J., Finger, T.D., Garfin, G. & Rattray, N. (2006). Drought and declining reservoirs: Comparing media discourse in Arizona and New Mexico, 2002-2004. *Global Environmental Change 16*: 95–113.

Spinoni, J., Naumann, G. & Vogt, J.V. (2015). The biggest drought events in Europe from 1950 to 2012. *Journal of Hydrology: Regional Studies 3*: 509–524.

Stagge, J.H., Kohn, I., Tallaksen, L.M. & Stahl, K. (2015). Modeling drought impact occurrence based on meteorological drought indices in Europe. *Journal of Hydrology 530*: 37–50.

Stahl, K., Kohn, I., Blauhut, V., Urquijo Reguera, J., De Stefano, L., Acácio, V., … van Lanen, H.A.J. (2016). Impact of European drought events: international database of text-based reports. *Natural Hazards and Earth System Sciences 16*: 801–819.

Steinemann, A.C. & Cavalcanti, L.F. (2006). Developing multiple indicators and triggers for drought plans. *Journal of Water Resources Planning and Management 132*(3): 164–174.

Steinemann, A.C., Iacobellis, S.F. & Cayan, D.R. (2015). Developing and Evaluating Drought Indicators for Decision-Making. *Journal of Hydrometeorology 16*: 1793–1803.

Su, Z. (2002). The Surface Energy Balance System (SEBS) for estimation of turbulent heat fluxes. *Hydrology and Earth System Sciences 6*(1): 85–99.

Svoboda, M., LeComte, D., Hayes, M., Heim, R., Gleason, K., Angel, J., … others. (2002). The drought monitor. *Bulletin of the American Meteorological Society 83*(8): 1181–1190.

Tsakiris, G., Pangalou, D. & Vangelis, H. (2006). Regional Drought Assessment Based on the Reconnaissance Drought Index (RDI). *Water Resources Management 21*: 821–833.

UNISDR. (2009). *Drought Rist Reduction Framework and Practices: Contributing to the Implementation of the Hyogo Framework for Action*. Geneva, Switzerland: United Nations secretariat of the International Strategy for Disaster Reduction (UNISDR). Retrieved from http://www.unisdr.org/files/11541_DroughtRiskReduction2009library.pdf

van Dijk, A.I.J.. & Renzullo, L.J. (2011). Water resource monitoring system and the role of satellite observations. *Hydrology and Earth System Sciences 15*: 39–55.

Vasiliades, L., Loukas, A. & Liberis, N. (2011). A water balance derived drought index for Pinios River Basin, Greece. *Water resources management 25*(4): 1087–1101.

Verma, M., Friedl, M.A., Richardson, A.D., Kiely, G., Cescatti, A., Law, B.E., … Propastin, P. (2014). Remote sensing of annual terrestrial gross primary productivity from MODIS: an assessment using the FLUXNET La Thuile data set. *Bogeosciences 11*: 2185–2200.

Vicente-Serrano, S.M., Beguería, S., Lorenzo Lacruz, J., Camarero, J.J., López-Moreno, J.I., Azorin-Molina, C., … Sanchez-Lorenzo, A. (2012). Performance of Drought Indices for Ecological, Agricultural, and Hydrological Applications. *Earth Interactions 16*: 27.

Vicente-Serrano, S.M., Cuadrat-Prats, J.M. & Romo, A. (2006). Early prediction of crop production using drought indices at different time-scales and remote sensing data: application in the Ebro Valley (north-east Spain). *International Journal of Remote Sensing 27*(3): 511–518.

Wagner, W., Dorigo, W., De Jeu, R., Fernandez, D., Benveniste, J., Haas, E. & Ertl, M. (2012). Fusion of active and passive microwave observations to create an essential climate variable data record on soil moisture. In *ISPRS Annals of the Photogrammetry, Remote Sensing and Spatial Information Sciences* (Vol. I-7). Presented at the XXII ISPRS Congress, Melbourne.

Wilhite, D.A. (2000). Drought as a Natural Hazard: Concepts and Definitions. In D. A. Wilhite (ed.), *Drought: A Global Assessment* (Vol. Vol. I). London: Routledge.

Wilhite, D.A. (2011). National Drought Policies: Addressing Impacts and Societal Vulnerability. In M. V. K. Sivakumar, R. P. Motha, D. A. Wilhile & J. J. Qu (eds.), *Towards a Compendium on National Drought Policies: Proceedings of an Expert Meeting, July 14-15, 2011, Washington, D.C. USA*. Geneva, Switzerland: World Meteorological Organization. Retrieved from http://digitalcommons.unl.edu/cgi/viewcontent.cgi?article=1079&context=droughtfacpub

Wilhite, D.A., Svoboda, M.D. & Hayes, M.J. (2007). Understanding the complex impacts of drought: a key to enhancing drought mitigation and preparedness. *Water Resources Management 21*(5): 763–774.

Wilhite, D.A. & Vanyarkho, O. (2000). Drought: Pervasive Impacts of a Creeping Phenomenon. In D. A. Wilhite (ed.), *Drought: A Global Assessment* (Vol. I). London: Routledge.

World Meteorological Organization (WMO). (2012). *Standardized Precipitation Index - User guide*. Geneva, Switzerland: World Meteorological Organization. Retrieved from http://www.wamis.org/agm/pubs/SPI/WMO_1090_EN.pdf

Zhang, F., Chen, J.M., Chen, J., Gough, C.M., Martin, T.A. & Dragoni, D. (2012). Evaluating spatial and temporal patterns of MODIS GPP over the conterminous U.S. against flux measurements and a process model. *Remote Sensing of Environment 2012*: 717–729.